



# Columnar and surface urban aerosol in Moscow megacity according to measurements and simulations with COSMO-ART model

Natalia E. Chubarova[1], Elizaveta E. Androsova[1], Alexander A. Kirsanov[2], Olga B. Popovicheva[3], Bernhard Vogel[4], Heike Vogel[4], Gdaliy S. Rivin[2]

[1] Faculty of Geography, Lomonosov Moscow State University, Moscow, 119991, Russian Federation
[2] Hydrometeorological Research Center of Russian Federation, Moscow, 123242, Russian Federation
[3] Faculty of Physics, Lomonosov Moscow State University, Moscow, 119991, Russian Federation
[4] Karlsruhe Institute of Technology, Karlsruhe, Germany

*Correspondence to*: Natalia E. Chubarova (natalia.chubarova@gmail.com)

**Abstract.** Urban aerosol pollution was analyzed over the Moscow megacity region using COSMO-ART chemical transport model and intensive measurement campaigns at the Moscow State University Meteorological Observatory (MSU MO, 55.707°N, 37.522°E) during April-May period in 2018 and 2019. We analyzed mass concentrations of Particulate Matter with diameter smaller 10 μm ($PM_{10}$), Black Carbon (BC), and aerosol gas precursors (NOx, $SO_2$, CHx) as well as columnar aerosol parameters for fine and coarse modes together with different meteorological parameters including an index characterizing the

Intensity of Particle Dispersion (IPD). Both model and experimental datasets have shown a statistically significant linear correlation of BC with $NO_2$ and $PM_{10}$ mass concentrations, which indicates mostly common sources of emissions of these substances. There was a pronounced increase in the $BC/PM_{10}$ ratio from 0.7% to 5.9% with the decrease in IPD index related to the amplification of the atmospheric stratification. We also found an inverse dependence between the $BC/PM_{10}$ ratio and columnar single scattering albedo (SSA) for the intense air mixing conditions. This dependence together with the obtained

negative correlation between wind speed and $BC/PM_{10}$ may serve an indicator of changes in the absorbing properties of the atmosphere due to meteorological factors. On average, relatively low for urban regions $BC/PM_{10}$ ratio of 4.7% is the cause of the observed relatively high SSA=0.94 in Moscow. Using long-term parallel aerosol optical depth (AOD) measurements over the 2006-2020 period at the MSU MO and in upwind clean background conditions at Zvenigorod Scientific Station (ZSS) of the IAP RAS (55.7N, 36.8E), we estimated the urban component of AOD ($AOD_{urb}$) and some other parameters as the

differences at these sites. The average $AOD_{urb}$ at 550nm was about 0.021 with more than 85% of fine aerosol mode. The comparisons between $AOD_{urb}$ obtained from model and measurements during the experiment have revealed a similar level of aerosol pollution of about $AOD_{urb}$ = 0.015-0.019, which comprised 15-19% of the total AOD at 550nm. The urban component of $PM_{10}$ ($PM_{10urb}$) was about 0.016 $mg\,m^{-3}$ according to the measurements and 0.006 $mg\,m^{-3}$ according to the COSMO-ART simulations. We obtained a pronounced diurnal cycle of $PM_{10urb}$ and urban BC ($BC_{urb}$), as well as their strong correlation with

the IPDs. With the IPD index change from 3 to 1 at night, there was about 4 times increase in $PM_{10urb}$ (up to 0.030-0.040 $mg\,m^{-3}$) and 3 times increase in $BC_{urb}$ (up to 0.003-0.0035 $mg\,m^{-3}$). At the same time, no pronounced daily cycle was found for the columnar urban aerosol component ($AOD_{urb}$), although there is a slight tendency to the increase in model $AOD_{urb}$ at night. We also obtained a close relationship between the calculated and measured $PM_{10urb}$ values, their dependence on IPD index, and the pronounced growth of $PM_{10urb}$ with the $PM_{10}$ increase.

## 1. Introduction

Anthropogenic aerosol pollution has a complex impact on the atmosphere, significantly affecting solar radiation, air temperature and humidity, and resulting in noticeable climatic effects (IPCC, 2021, IPCC, 2013; Jacobson, 2004; Bond et al., 2013). Aerosol particles at surface level also have a harmful effect on human health (Manisalidis et al., 2020, Lu et al., 2015).



Radiative effects of anthropogenic aerosol exceed 1 W / m2 in absolute value, partially compensating the increase in air
temperature in the troposphere due to rising the concentration of greenhouse gases; however, the uncertainty of aerosol climate
impact estimates remains quite high (IPCC, 2021, Myhre et al., 2013). These uncertainties are associated with a wide variety
of optical and microphysical characteristics of aerosol (Seinfeld Pandis, 2016) and its significant temporal and spatial variation.
Anthropogenic aerosol is considered to be smaller in size and is more absorbing than natural aerosol (Myhre, 2009; Su et al.,
2013, Kinne et al, 2013).

To date, these features of the urban aerosol pollution have not been fully studied, despite the significant efforts of scientific
community and the existence of  different international aerosol programs within the World Data Centre for Aerosols,
https://gaw-wdca.org): AERONET (Aerosol Robotic Network, https://aeronet.gsfc.nasa.gov/), ACTRIS (Aerosol, Clouds and
Trace Gases,   https://actris.nilu.no),    AEROCOM (Aerosol Comparisons between Observations and Models,
https://aerocom.met.no/)).

The intensive aerosol studies concern the optical properties of urban aerosol, its relation with meteorological characteristics as
well as the relationship between surface concentration and columnar aerosol content and the emission sources (Segura et al.,
2017, Zhuang et al., 2018, Wang et al., 2019, Zhdanova et al., 2020). However, in most publications the authors consider both
natural and urban aerosol columnar aerosol in polluted areas without highlighting its urban component (Kumar et at., 2019,
Chou et al., 2006, Zhuang et al., 2018, Segura et al., 2017). In only few papers the urban component of columnar aerosol
optical depth (AOD) and other aerosol properties were evaluated (Zawadska et al., 2013, Chubarova et al., 2011, Zhdanova et
al., 2020). At the same time, the detection of urban aerosol component and its relationship with anthropogenic emissions of
gas precursors are critical for assessing aerosol radiation forcing and its climate effect (IPCC, 2021, Remer and Kaufman,
1998).

For better testing urban aerosol and its relationship with meteorological conditions it is important to analyze the links between
surface and columnar aerosol content.  However, such kind of the analysis was performed only in few publications (Segura et
al., 2017, Wang et al., 2019, Gubanova et al., 2018).

A particularly important urban component is black carbon (BC), which absorbs visible radiation and contributes to the heating
of the atmosphere contrary to most other aerosol species (Bond et al. 2013; Jacobson, 2004, 2006, Ramanathan and Carmichael,
2008). The urban environment is the main source of black carbon emissions due to the use of diesel fuel (Weingartner et al.,
1997). The contribution of emissions from heavy vehicles (trucks, buses, etc.) with diesel engines can reach 42% of the total
mass of black carbon emissions into the atmosphere (Reddy and Venkataraman, 2002). There is still the lack of information
on BC measurements even at surface level, which provides a gap in understanding the balance between heating and cooling
rates in the atmosphere (Bond et al. 2013), especially in large cities, where BC emissions are high. In addition, possible
influence of BC on absorbing properties of the whole atmosphere, which is important for climate effect evaluation, have been
analyzed only in few publications (Markowicz et al., 2017, Rajesh et al., 2018, Kozlov et al., 2016).  Therefore, the model
assessment of BC at urban sites and its testing against measurements are the important tasks (Gilardoni et al., 2011, Lugon et
al., 2021, Tang et al.,2021).

The importance of aerosol research and specifying the urban aerosol is also associated with the need to improve the accuracy
of the forecast of meteorological characteristics, which noticeably depends on aerosol amount and its properties (Huang and
Ding, 2021, Toll et al., 2016, Poliukhov and Blinov, 2021, Wang et al., 2020). The analysis of the aerosol pollution and its
influence on meteorological regime are usually performed with the help of chemical transport models (CTM) coupled with
weather prediction models or the data of re-analysis (Baklanov et al., 2017; Evans et al., 2003; Vogel et al., 2010; WMO-
COST, 2008). However, to obtain reliable estimates of aerosol pollution, careful testing of the simulations against the results
of measurement campaigns is required. For example, in (Ukhov et al., 2020) the application of the WRF-Chem model over



Middle East and its testing against measurements provides the reliable assessment of the pollution by mineral and sulphate aerosol over the urban area in this region.

According to AEROCOM modeling data and CMIP5 model assessments, the anthropogenic component of AOD at a wavelength of 550 nm is $0.03 \pm 0.01$, which is $24 \pm 6\%$ of the total AOD (IPCC, 2013). This is smaller than that obtained by satellite measurements, which according to (Loeb and Su, 2010, Bellouin et al.,2013) provide the estimates of about 0.06 over

land, comprising about 20-40% of the total AOD. According to the latest estimates there is still an underestimation in simulated aerosol optical depth (AOD) of about 21 % against measurements (Gliß et al., 2021). Hence, the evaluation of urban aerosol component and its ratio in total AOD from measurements may provide a helpful testbed for aerosol urban modelling.

Moscow megacity with its population of about 13 million of people and with about 7 million of vehicles is one of the largest urban agglomeration in the world. As a capital of Russian Federation, it is a large financial and administrative center.

According to air pollution, Moscow is among slightly pollutant megacities in Europe and North America (Elansky et al. 2014; Elansky et al., 2018). BC measurements in the Moscow center have revealed the level of air pollution, which is substantially lower than in Beijing (Golitsyn et al., 2015). Transport - related BC in spring of 2018 and 2019 in Moscow urban background was found comparable to Helsinki, least polluted city in Europe (Popovicheva et al., 2020 a,b).

The main tasks of this paper concern the analysis of the aerosol properties at surface and in the atmospheric column, their

relationship with meteorological parameters, and the dynamics of aerosol in the urban environment of the Moscow megacity using the results of the chemical transport model and the data obtained during the intensive measurement campaigns over Moscow region in spring periods of 2018 and 2019.

## 2. The methods

### 2.1 The description of measurements and model experiments

For a detailed study of the properties of atmospheric aerosol and its urban component a complex experiment has been organized. It consisted of the intensive measurement campaigns at the Meteorological Observatory of Moscow State University (MSU MO), located at 55.7°N, 37.52°E (Fig. 1), and model simulations using the Russian COSMO-Ru-ART (COSMO — COnsortium for Small-scale MOdelling, ART — Aerosols and Reactive Trace gases) configuration (Vil'fand et al., 2017) of the COSMO-ART model system (Vogel et al., 2010) over the whole Moscow area and surrounding territories. The

measurement campaign and model simulations covered the periods of April- May, 2018 and 2019.

#### 2.1.1. Measurements

The MSU MO is located on the territory of the MSU Botanical Garden in the park area at a distance of several km from the local sources of emissions (power stations). The nearest highways are about 300–450 m away from the site. During the intensive measurement campaigns, the mass concentrations of various gas-aerosol species at the Earth's surface and aerosol

characteristics in the total atmospheric column were studied together with meteorological observations. The measurements comprised both surface aerosol properties, including mass concentrations of particulate matter with diameter smaller 10 μm ($PM_{10}$) and Black Carbon (BC), different aerosol gas precursors (NOx, $SO_2$, CHx), other gas species (O3, CO), and the columnar aerosol characteristics according to the AERONET data.

The mass concentration of $PM_{10}$, the aerosol precursor and other gas species (NO, $NO_2$, $SO_2$, volatile organic compounds

VOCs, marked as CHx, and CO) measurements were carried out with a 20-minute time resolution. These observations have been in operation by the Mosecomonitoring State Environmental Protection Agency. The TEOM 1400a (Thermo Environmental Instruments Inc., USA) was used for $PM_{10}$ measurements. Internationally certified OPTEC Russian instruments (www.optec.ru) were applied to measure gas species concentrations of NOx, $O_3$, $SO_2$, and CO. The Gamma-ET instrument





(http://etek-ltd.ru) was used for the CHx measurements. The description of the quality assurance (QA) procedures is given at http://mosecom.mos.ru.

Aerosol equivalent BC (eBC) mass concentrations were measured using custom-made portable aethalometer. In this instrument the light attenuation caused by the particles depositing on a quartz fiber was analyzed at three wavelengths (450, 550, and 650 nm). The eBC concentrations were determined by converting the time-resolved light attenuation to eBC mass at 650 nm and characterized by a specific mean mass attenuation coefficient, as described in (Popovicheva et al., 2017). Calibration parameter for quantification eBC mass was derived during parallel long - term measurements against an AE33 aethalometer (Magee Scientific) that operates at the same three wavelengths, for more details see elsewhere (Popovicheva et al., 2020a).

Aerosol measurements in the atmospheric column were carried out using the CIMEL sun/sky photometer, which is operated at the Moscow State University in the framework of the AERONET program since 2001 (Holben et al. 1998; Chubarova et al., 2011a). The following columnar aerosol characteristics were analyzed: aerosol optical depth (AOD) in the spectral range from 340 nm to 1020 nm, AOD fine and coarse modes at a wavelength of 500 nm (O'Neill et al., 2001), the Angstrom extinction exponent (AEE) and the Angstrom absorption exponent (AAE) in the spectral range of 440-870 nm, single scattering albedo (SSA) at 675 nm, asymmetry factor at 675 nm for various aerosol modes calculated in accordance with the AERONET algorithms (Dubovik and King, 2000). We used the latest version 3 AERONET dataset at level 2 (Giles et al., 2019) with 15- minute resolution during daytime. A detailed testing of these data has confirmed the results obtained in (Giles et al., 2019), that the new algorithm of automatic cloud filtration worked much better than the previous AERONET algorithm in the version 2 with the exception of winter months (Aerosol urban pollution..., 2020). As a result, we did not apply an additional cloud filtering as it has been previously done (Chubarova et al., 2016). For comparing the measured and model values of aerosol optical depth, the AOD obtained from measurements was recalculated to a wavelength of 550 nm from AOD at 500 nm (AOD500) using the AEE parameter.

We also used meteorological observations (air temperature, atmospheric pressure, wind speed, wind direction) with 1-minute resolution from the Vaisala MAWS-301 automatic weather station, as well as standard meteorological MSU MO measurements with 3- hour resolution. For characterizing meteorological conditions, which influence on the dynamics of air pollution, we used the index characterizing the Intensity of Particle Dispersion (IPD) described in (Kuznetsova et al., 2014). It is calculated using a set of meteorological parameters including atmospheric pressure conditions, the type of atmospheric circulation, the stratification of the atmosphere, wind speed up to 850 hPa and level of precipitation. IPD index varies from 1 to 3. The conditions with IPD= 1 are characterized by a stable stratification of the atmosphere, a low-gradient baric field, low wind speed conditions, and the absence of precipitation. At IPD=3 the opposite picture is observed with intensive air mixing conditions, high wind speed, precipitation, unstable stratification, frontal zones. In our study the IPD index is evaluated using the 24-hour COSMO mesoscale model forecast with 1-hour resolution.

### 2.1.2. COSMO-ART model and numerical experiments

The urban aerosol characteristics were calculated using the COSMO-Ru-ART model system (Vogel et al. 2010, Vil'fand et al., 2017, Rivin et al., 2019) with horizontal grid step of 7 km and time resolution of 40 s over the 1000x1000 km area. In this model system meteorological simulations are performed by the COSMO mesoscale model (http://www.cosmo-model.org/), an operational weather prediction model at the Russian Hydrometeorological Centre (Rivin et al., 2019). The gas-aerosol concentrations are simulated using the ART chemical transport model, which is coupled with the COSMO model. The COSMO-Ru-ART system reproduces chemical transformations of substances in the gas phase and heterogeneous reactions, photolysis, nucleation, coagulation, condensation, emissions of various types of aerosols, dry and wet aerosol deposition (Vogel et al. 2010, Vil'fand et al., 2017). About 172 chemical reactions are used in the ART model to describe chemical processes in the troposphere. One of the most important features of COSMO-Ru-ART is the parallel calculation of




meteorological parameters and chemical transformations at each time step, which allows a user to take into account the reverse effect of aerosols on radiation and meteorological characteristics of the atmosphere. As a result, using the COSMO-Ru-ART model system we can quantify the rate of formation of new aerosol particles and aerosol gas precursors in the polluted urban atmosphere in real atmospheric conditions, which in turn are modified by the updated chemical composition. Its full description is given in (Vil'fand et al., 2017). The data of the forecast of the COSMO-Ru system (Rivin et al., 2019) and the global ICON

model were used as initial and boundary conditions.

In addition, for the operation of the ART model, the data from the Global Land Cover 2000 project on land use and inventory data from TNO2010 (the Netherlands Organization for Applied Scientific Research, https://www.tno.nl/en/) were applied to determine anthropogenic emissions of pollutants. The TNO2010 emission inventory has been developed using official reported emissions data by source category and combining them with other estimates where needed (Kuenen et al., 2014). The spatial

distribution of monthly mean aerosol gas precursor and $PM_{10}$ emissions over Moscow area for April and May is shown in Fig. 2. One can see that most of the urban emissions are observed over the center of Moscow megacity due to the influence of traffic. In April the emissions of $SO_2$ and $NO_x$ are larger due to heating season. Aerosol concentrations at the borders of the simulated area were assumed to be close to zero to exclude the influence of regional background aerosol and aerosol gas precursor effects. So the simulated gas and aerosol concentrations are associated only with the urban Moscow emissions. The

time set for aerosol generation was equal or higher 31 hours in accordance with the recommendations of the model developers (Vogel, private communication). Thus, we consider the simulation of only the anthropogenic components of the surface mass concentrations of $PM_{10}$, BC, and gas aerosol precursors, as well as the columnar urban component of aerosol optical depth at 550 nm and single scattering albedo (SSA).

### 2.2. Evaluation of urban aerosol component

For identifying the urban component of aerosol, we compared the results of parallel measurements and model simulations over the MSU MO and Zvenigorod Scientific Station (ZSS) of the A. M. Obukhov Institute of Atmospheric Physics (IAP) (55.7°N, 36.8°E) located 55 km to the west of the MSU MO (see Fig. 1). Due to prevailing westerlies and location of the ZSS site far from local anthropogenic emissions (see Fig. 2) it can be characterized as a background site. This kind of diagnostics provides us the reliable estimates of urban aerosol effect over large Moscow megacity.

The urban component of columnar aerosol optical depth from both measurements and modelling was estimated as the difference between the data at the MSU MO (marked as Moscow) and at the background ZSS site (marked as Zven):

$$AOD_{urb}=AOD_{\ Moscow}\text{-}AOD_{\ Zven} \tag{1}$$

In similar way, we estimated the urban components of some other columnar aerosol parameters, such as AEE, fine and coarse mode of AOD at 500nm.

At ZSS the AERONET measurements have been in operation since 2006, therefore the parallel measurements between Moscow and Zvenigorod were analyzed for the 2006-2020 period. The time difference between the two instant measurements in these sites is only 3 minutes.

Similar approach was used for evaluating the urban component of $PM_{10}$ mass concentration:

$$PM_{10urb}=PM_{10\ Moscow}\ \text{-}\ PM_{10\ Zven} \tag{2}$$

In Zvenigorod the $PM_{10}$ mass concentration was also measured with the help of the TEOM 1400a instrument by the Mosecomonitoring Agency. Since the data were available only for 2018, the comparisons of $PM_{10}$ urban component was made only for this year.

We consider that our BC measurements in Moscow provide the $BC_{urb}$ component, whereas the black carbon is mainly formed and emitted in the urban environment (see Fig. 2).





The joint use of both measurements and modelling of aerosol properties of the atmosphere in Moscow region provides more reliable assessment of urban aerosol pollution.

For the evaluation of the urban aerosol component in Moscow and for the accurate comparisons of model calculations with measurements it was necessary to remove the cases with the influence of smoke aerosol, which has different optical properties (Dubovik et al., 2002; Liu et al., 2018). For the initial information on location of biomass burning event we used FIRMS (FIRe

Monitoring Service) dataset (https://firms.modaps.eosdis.nasa.gov/). After identification of biomass burning spots we applied the backward trajectory analysis using the READY system (Rolph et al., 2017) with the help of HYSPLIT visualization model at the height of 0.5-3 km (Stein et al., 2015). We consider that the air mass is affected by biomass burning aerosol if the cases are detected within 50 km from the line of particle motion. If number of biomass burning spots were smaller than 5, in addition, we analyzed Angstrom absorption exponent (AAE) measurements from AERONET at the MSU MO and used the threshold

of AAE<1 to reveal typical aerosol for Moscow area. During the low-temperature biomass burning process the AAE values should be higher than 1, because of much intensive absorption at shorter wavelengths (at 440nm in our case) by organic carbon (Kirchstetter et al., 2004; Sun et al., 2017). More details of this method can be found in (Chubarova et al., 2021).

It should be noted that this procedure has been applied only for the comparisons between model and measured aerosol parameters, since no fire emissions have been accounted for in the model simulations.

**3. Results**

**3.1. Aerosol characteristics in Moscow according to long-term AERONET measurements**

In order to understand whether or not the aerosol features during the intensive experiment were representative for the whole warm period we analyzed the results of long-term aerosol measurements using the MSU MO AERONET dataset from 2001 to 2020. Figure 3 presents seasonal variability of the AOD at 500nm (AOD500), its fine and coarse modes and Angstrom

extinction exponent (AEE) according to long-term observations and, in particular, for April and May in 2018 and 2019. One can see a noticeable AOD500 increase during warm period. The spring maximum of AOD500 is associated with the descent of snow cover, and the effects of seasonal agricultural biomass burning in conditions with low precipitation typical for this period (Chubarova et al., 2014). The elevated spring AOD500 values are accompanied by lower AEE, which also is in an agreement with slightly smaller fraction of the fine mode AOD500. The summer AOD500 maximum is associated with the

active formation of submicron aerosol with fine mode AOD500 fraction higher 80%. The April-May period of 2018-2019 is characterized by slightly lower AOD500, which is in an agreement with a negative AOD500 trend in Moscow in recent years (Zhdanova et al.,2020; Chubarova et al., 2016). The lower fraction of fine mode aerosol (64% compared to 71-73%) may also indicate the decrease in the formation of secondary aerosol due to the effective reduction of urban gas precursor emissions in Moscow (Zhdanova et al., 2020). However, in general, the aerosol conditions in April and May of 2018-2019 correspond to

those during warm period with a slightly reduced AOD500 and its fine mode fraction, which corresponds to the observed trends of purification of the Moscow atmosphere in recent years.

**3.2. Main characteristics of aerosol and aerosol gas precursors, and their relationship according to the intensive measurements campaigns of 2018-2019**

Table 1 shows the statistics of aerosol and gas parameters of the atmosphere during the spring intensive measurement

campaigns in 2018 and 2019. Median value of AOD at 500 nm was small (0.12), corresponding to its level in Central and Northern Europe (Chubarova, 2009; Filonchuk etc., 2019) with predominance of fine mode aerosol. Median $PM_{10}$ value of 0.025 mg m$^{-3}$ is also relatively small and is significantly lower than the $PM_{10}$ concentrations in Chinese megacities, where





average concentrations exceed $0.1 \, \mathrm{mg\,m^{-3}}$ (Climate of Moscow, 2017). However, for some days (April 16, 2018, April 22, 25 and 27, 2019), we observed an elevated $PM_{10}$ levels exceeding the threshold of daily maximum allowable concentration of

$0.060 \, \mathrm{mg\,m^{-3}}$ adopted as Russian standard. The median SSA of 0.94 is typical for slightly absorbing aerosol, which is in agreement with rather low $BC/PM_{10}$ ratio (4.3%) and relatively low mean concentrations of BC ($1.03 \, \mathrm{\mu g\,m^{-3}}$). Note, that BC concentrations are only $0.4$-$0.5 \, \mathrm{\mu g\,m^{-3}}$ over remote areas in Paerne (Switzerland) and Reunion Island (France) according to (Gerich et al., 2011, Bhugwant and Brémaud, 2001). At the same time, in some conditions in Moscow an increase in hourly BC up to $8.9 \, \mathrm{\mu g\,m^{-3}}$ was observed. This corresponds to high BC concentrations varying from $5.5 \, \mathrm{\mu g\,m^{-3}}$ in Dhanbad (India)

(Singh et al., 2015) up to $9 \, \mathrm{\mu g\,m^{-3}}$ in Guangzhou (China) (Wu et al., 2013). Due to the predominance of fine mode aerosol, the asymmetry factor of the aerosol phase function is relatively small (about 0.63 if considering both fine and coarse aerosol modes), which also corresponds to relatively high AEE values (Dubovik et al., 2002).

The analysis of the aerosol gas precursors revealed very low concentrations of sulfur dioxide in Moscow, while nitrogen oxides are traditionally high due to the strong traffic in the city and emissions from power plants (Report on the state of environment

in Moscow, 2020).

Figure 4 presents the time series of daily mean AOD at 500 nm, $PM_{10}$, $BC/PM_{10}$ as well as the concentrations of the main aerosol gas precursors during the intensive campaigns. For characterizing meteorological conditions we also show daily variability of water vapor content W and the IPD indices. There are large variations in both surface and columnar aerosol characteristics of the atmosphere during these periods. In the stable atmosphere with daily mean IPD of about 2 an elevated

columnar and surface aerosol loadings are observed (for example, on April 12-16, 2018, May 14-16, 2018, April 20-22, 2019). However, during the days affected by the advection of biomass burning aerosol (for example, 1.05.2018, 27.04.2019), there is high aerosol loading even in good air mixing conditions at IPD=3. These days were also characterized by the elevated NOx concentrations due to the active chemical transformation affected by forest fires (Jin et al., 2021). Note, that high NOx level is observed, in spite of low traffic due to weekend (27.04.2019) or holiday (1.05.2018).

A correlation matrix has been estimated for evaluating the relationship between different columnar and surface aerosol characteristics, aerosol gas precursors and meteorological parameters (Table 2). We obtained a statistically significant correlation of columnar AOD500 with surface $PM_{10}$, and BC. A more pronounced dependence of both BC and $PM_{10}$ with fine AOD500 mode could be explained by the fine mode BC composition and the predominant fraction of fine aerosol mode in $PM_{10}$ in urban aerosol in Central and Northern Europe (see, for example, see Fig. 10 in Wu and Boor, 2021). The importance

of secondary urban aerosol in columnar fine mode AOD500 (Dubovik et al., 2002) has been also proved by a statistically significant correlation between fine AOD500 mode and aerosol gas precursors ($NO_2$, $SO_2$, CHx), which indicates the importance of these substances for aerosol formation.

A positive correlation between water vapor content W in the atmospheric column and all aerosol parameters has revealed more favorable processes of aerosol formation in relatively warmer and wetter air masses. In addition, the advection of cold air

masses with small W and aerosol loading from northern regions may be also the important cause of this correlation (Szkop A. et al., 2016). There are also high negative correlations of AOD with surface wind speed due to ventilation effect in the urban environment, which occurs due to blowing the urban aerosol out of Moscow. A decrease in AEE and, correspondingly, the decrease in the fine AOD fraction with the increase in wind speed may be also associated with less effective fine mode aerosol generation due to better ventilation conditions. This is also in accordance with statistically significant correlation between

wind speed and aerosol gas precursors. The exception is sulfur dioxide, which concentrations are extremely small in Moscow (see Table 1), and therefore large errors can be observed, when detecting these relations. The pronounced negative correlations with wind speed were found for surface aerosol species such as $PM_{10}$, and, especially, BC. Negative correlation between BC and wind speed was also shown in (Popovicheva et al. 2020a, Chen et al., 2014). Note that the observed negative correlation


of BC/PM$_{10}$ ratio with wind speed may lead to the decrease in the absorbing properties of the atmosphere in case of high wind speed.

A statistically significant correlation between surface aerosol gas precursors and IPD index confirms that at higher IPD there are better conditions for intense air mixing, which, as a result, provide a decrease in aerosol gas precursor mass concentration. However, the correlation of columnar AOD and PM$_{10}$ with IPD index, contrary to wind speed, is not statistically significant, probably due to the prevailing effects of natural aerosol in AOD and PM$_{10}$. The closer relationship of wind speed and IPD with BC compared to PM$_{10}$ indicates more important role of local meteorological situation for black carbon, since urban emissions of pollutants is the main source of BC, while for PM$_{10}$, in addition, we have a regional aerosol source, which undergo significant variations (Air quality in Europe, 2020).

A more detailed analysis of the relationship between AOD500 and PM$_{10}$ surface mass concentrations shown in Fig. 5a demonstrates that along with the existence of a general dependence, there is a split into two types at a point of bifurcation of PM$_{10}$ ~0.05 mg m$^{-3}$. A weaker AOD500 dependence versus PM$_{10}$ characterizes the accumulation of PM$_{10}$ only in the low layer (due to local emission sources near the surface) in the absence of the pronounced AOD increase with many cases of IPD=1, relating to the low intensity of particle dispersion. A more pronounced dependence between AOD and PM$_{10}$ is associated with the influence of air mass advection, when the concentration of surface particles increases simultaneously with AOD. In this case only few cases of IPD=1 are observed (Fig. 5a). The increase in PM$_{10}$ is also connected with a significant increase in fine mode AOD fraction and the total absence of its low values at high PM$_{10}$ levels (Fig. 5b).

There are also noticeable variations in the BC/PM$_{10}$ ratio depending on PM$_{10}$ and IPD (Fig. 5c). In well mixing air conditions (IPD=3), much lower values of the BC/PM$_{10}$ ratio are observed: in most cases, they are smaller than 0.01 and decrease with the growth of PM$_{10}$. This corresponds to the situation, when there is an advection of air outside of Moscow, with high natural aerosol content, but with a relatively low BC content. On average, at IPD=3, the BC/PM$_{10}$ ratio is equal to 0.7%. At the same time, with the IPD decrease BC/PM$_{10}$ ratio is getting higher with mean value of 5.5% and 5.9%, respectively, for IPD=2 and IPD=1. Thus, the use of IPD data may significantly refine the BC/PM$_{10}$ level and, as a result, the absorbing properties of the atmosphere.

Figure 6 presents the scattering diagrams of BC mass concentration as a function of PM$_{10}$, NO$_2$, and SO$_2$ for different IPD regimes obtained according to both measurements and COSMO-ART simulations. Model simulations confirm close relationships of BC with PM$_{10}$ and NO$_2$. At the same time, the correlation of BC with sulfur dioxide was revealed only by modeling at relatively high concentrations of SO$_2$, which are not observed in Moscow (Report on the state of environment in Moscow, 2019; Climate of Moscow.., 2017). This indicates that the data on SO$_2$ emissions in Moscow according to TNO2010 were overestimated. The main source of SO$_2$ emissions is usually the coal fuel at power plants, which practically is not used in the Moscow region, except in situations of extremely cold winter (Climate of Moscow.., 2017).

Since black carbon is an important aerosol component, which strongly absorbs visible radiation, and its measurements are very sparse, in some cases it may be necessary to evaluate its concentration according to the available measurements of the gas composition at environmental monitoring stations. According to our measurements, hourly values of the BC mass concentration (in μg m$^{-3}$) can be evaluated from PM$_{10}$ (in μg m$^{-3}$) or NO$_2$ (μg m$^{-3}$) using the following regression equations:

$$BC = 0.036\,PM_{10} + 0.111,\ R = 0.64 \tag{3}$$

$$BC = 0.035\,NO_2 + 0.174,\ \ R = 0.70 \tag{4}$$

where $R$ is the Pearson correlation coefficient.

These regression dependences can be used as a first approximation for the estimates of BC concentrations for the warm period with relatively high temperatures and high solar elevations providing favorable conditions for photochemistry, which is important for NO$_2$ production. The close results obtained by modelling confirmed the possibility of using these regression dependences.



### 3.3. Relationships between aerosol single scattering albedo and BC/PM$_{10}$ ratio

We noted earlier, that the BC/PM$_{10}$ ratio may characterize the absorbing properties of the aerosol. This is especially evident in the visible spectral range, where BC is almost the only source for the solar radiation absorption, and its high concentrations can lead to a decrease in aerosol single scattering albedo (Kozlov et al., 2008) and to the significant radiative effects. As a result, we propose to use BC/PM$_{10}$ ratio as first approximation for estimating the SSA values. The use of BC/PM$_{10}$ ratio might be also useful in different atmospheric tasks, since in the standard AERONET algorithm there is a strong limitation of SSA retrievals only for cloud-free conditions and relatively high aerosol loading (Dubovik and King, 2000). However, it is necessary to take into account, how accurately BC/PM$_{10}$ ratio at surface captures the conditions of the entire column of the atmosphere. According to our observations the restriction on IPD=3 is not enough for obtaining the relationship between them, and more strict conditions are required. Currently, we applied the limitation on daytime period (±3 hour around the solar noon), when a significant increase in air convection is observed during the warm period. The application of this additional restriction provides the dependence between SSA and BC/PM$_{10}$ (Fig. 7), which is close to the results obtained in previous experiment in Moscow (Chubarova et al., 2013). The dependence is not strong possibly due to the large uncertainty (about 0.03) of the SSA AERONET retrievals (Dubovik and King, 2000) and relatively small statistics. Model estimates of SSA dependence on BC/PM$_{10}$ ratio provide much more significant relationship with a correlation coefficient R=0.87, but the values themselves are lower and the SSA sensitivity to the BC/PM$_{10}$ value is higher. Thus, further analysis with more statistics is required for better attributing this dependence.

### 3.4. Aerosol urban pollution based on comparisons between Moscow and background conditions at the ZSS.

As described in Section 2, we estimated the urban aerosol pollution in Moscow megacity as the difference of aerosol characteristics between Moscow MSU MO and Zvenigorod site (see Eq. (1) and Eq.(2)). Figure 8 shows mean total urban component of AOD, fine and coarse mode of AOD$_{urb}$, and the urban component of AEE for the entire period of parallel AERONET observations in Moscow and Zvenigorod from 2006 to 2020. On average, AOD$_{urb}$ at 500nm was about 0.025 with prevailing fine_mode_AOD$_{urb}$=0.021, which is in agreement with the positive sign of AEE$_{urb}$. No statistically significant difference in coarse AOD mode between Moscow and clean unpolluted site was found. The inset in Fig. 8 shows the AOD$_{urb}$ spectral dependence, which is characterized by larger values at shorter wavelengths corresponding to urban fine mode aerosol. The inverse dependence of AOD in UV region with smaller AOD$_{urb}$ at 340 nm is due to a slight underestimation of the nitrogen dioxide content in the atmospheric column in Moscow, which is used in AOD retrievals. This underestimation was much larger in the version 2.0 of the AERONET dataset (see the discussion in Chubarova et al. (2011b)).

Over the April-May 2018-2019 period the detailed AOD$_{urb}$ model calculations were compared with the measured AOD$_{urb}$ for the cases without smoke air advection from the areas of forest and agricultural fires. In addition, we put a filter on cloud amount N<5 to make the comparison only for semi-clear sky conditions. The latter filtering is necessary, since in these conditions, we avoid the problems with too active generation of aerosol in cloudy conditions with high relative humidity in the COSMO-ART model system ("Aerosol urban pollution", 2020).

Figure 9a shows the time series of the measured and model components of AOD$_{urb}$ at a wavelength of 550 nm and the observed AOD in Moscow for April-May 2018-2019 period. The model AOD$_{urb}$ values vary mainly in the range of 0.05, reaching in some cases 0.1-0.17. The measured AOD$_{urb}$ varies in the larger range: from -0.12 to +0.14. Negative AOD$_{urb}$ may be associated with the influence of the advection of polluted air from Moscow, which will be analyzed later. On average, model and measured AOD$_{urb}$ comprises 0.015 and 0.019, respectively, being in satisfactory agreement (Table 3). Mean AOD$_{urb}$/AOD ratio from measurements comprises about 19%, and the model AOD$_{urb}$ is slightly smaller (15%). However, in conditions with relatively low AOD, the AOD$_{urb}$/AOD ratio can reach 58% (for example, on 21.05.2018).





Figure 9b presents the time series of the $PM_{10}$ mass concentration, and its model and measured urban components. One can see large variations of $PM_{10urb}$, especially according to the measurements, which can be negative. Note, that these negative $PM_{10urb}$ were observed only during the night or early in the morning. On average, model $PM_{10urb}$ is lower than the measured $PM_{10urb}$ values (0.006 and 0.016 mg m$^{-3}$, respectively (see Table 3)). Higher values of measured $PM_{10urb}$ provide larger

$PM_{10urb}/PM_{10}$ ratio of about 70%, while according to model estimates it is much smaller (about 27%). This may happen due to some underestimation of urban aerosol and gas emissions in Moscow megacity conditions, which should be studied further. Since BC is almost purely urban aerosol component in the absence of smoke aerosol advection, the model $BC_{urb}/BC$ ratio comprises more than 93% of the total BC. We also see slightly overestimated model BC compared with measurements (1.6 µg m$^{-3}$ compared with 0.95 µg m$^{-3}$), which also may results in too low model single-scattering albedo in urban conditions

shown in Fig. 7. This, in turn, may happen due to some overestimating of the BC emissions in the TNO2010 inventory dataset. The formation of both natural and urban aerosol depends on both chemical composition of the atmosphere and meteorological conditions. We analyzed if there is a relationship between urban aerosol component and the total aerosol content. Figure 10 presents the dependence of model and measured $AOD_{urb}$ on total AOD according to the MSU MO measurements, and the dependence of $PM_{10urb}$ on $PM_{10}$. There is a positive correlation of urban aerosol component for AOD and $PM_{10}$ with total AOD

and $PM_{10}$. This could be explained by more favorable meteorological and chemical conditions for generation both urban and natural aerosol. This may be also accompanied by higher concentrations of aerosol gas precursors both of urban and natural origin, which, in turn, have high correlations with $PM_{10}$ and AOD according to Table 2. At surface layer, for $PM_{10urb}$ the dependence is more pronounced, since the emissions of these substances are observed mainly close to ground. As for the dependence of measured $AOD_{urb}$ versus AOD at 550 nm (see Fig. 10a) variations are much larger due to the significant

contribution of natural aerosol component in observations.

As mentioned earlier, the increase in aerosol loading over Moscow suburbs can occur due to the advection of polluted air from Moscow. Therefore, the calculated and measured aerosol urban components were compared, in addition, for the cases, when the influence of the air advection from Moscow megacity was not observed. For removing such cases from the sample, we applied the HYSPLIT model ensembles of the 24-hour forward trajectories [Stein et al., 2015] at 500-1000m for noon

conditions. We consider that, since Zvenigorod site is located directly to the west of Moscow city center, the air quality was not affected by Moscow pollution if the trajectories were in the zone from 0 to 180 degrees. Figure 11 shows the dependence between model and measured $AOD_{urb}$ and $PM_{10urb}$ for all cases (Fig.11 a,b) and for the cases without air advection from Moscow (Fig.11 c,d). The analysis was made for the same AOD and $PM_{10}$ statistics obtained during daytime and in sunny conditions. We see that after the removal of the Moscow influence much fewer cases with the measured negative $AOD_{urb}$

values are observed (compare Fig. 11a and Fig.11c) and the remaining negative $AOD_{urb}$ do not exceed 0.01, which is the uncertainty of AOD measurements in AERONET (Holben et al., 1998). As a result, after removing of the Moscow affected cases mean value of $AOD_{urb}$ is equal to 0.019, which was only slightly higher compared with $AOD_{urb}$=0.016 obtained for all cases (see Table 3). Similarly Figure 11 b,d presents the relationships between calculated and measured $PM_{10urb}$ for all cases and for the cases without the Moscow influence. Interestingly, that during daytime there were no negative $PM_{10urb}$ values,

when $PM_{10\ zven} > PM_{10\ Moscow}$ as shown in Fig.9b. Both simulated and measured $PM_{10urb}$ values have a pronounced dependence on IPD with higher $PM_{10urb}$ at lower level of intensity of particle dispersion. Note, that the influence of the intensity of particle dispersion on $AOD_{urb}$ is not observed.

Since the emissions and the intensity of particle dispersion have a pronounced daily course, the urban aerosol component may also have significant differences. Figure 12 shows the daily cycles of AOD, $PM_{10}$ and BC, as well as the primary emissions of black carbon and $PM_{10}$ according to TNO2010 inventory. In general, there are consistent diurnal changes of model and

experimental data at the surface layer. One can see the accumulation of $PM_{10}$ and BC at night below the inversion layer in the stable atmosphere, which is characterized by IPD=1. During night time AOD measurements are not available, therefore $AOD_{urb}$





diurnal changes are evaluated only from morning to evening. One can see no evident dependence of measured $AOD_{urb}$ changes within this period, however, according to model estimates there is a small $AOD_{urb}$ increase at night, especially, in conditions

with IPD=1. Figure 12 also demonstrates strong dependence of BC level on IPD index, especially for night and early morning conditions with prevailing low intensity of particle dispersion. Elevated values of the surface urban aerosol at night in conditions with IPD=1 reach 30-40 $\mu g\,m^{-3}$ for $PM_{10urb}$, and to 3-3.5 $\mu g\,m^{-3}$ - for BC.

The BC diurnal cycle is mainly determined by variations of the boundary layer of the atmosphere. In warm period there is an increase of its height during daytime, which contributes to the processes of dilution and strengthening of convective processes

due to the additional heating by solar radiation, which leads to a decrease in the concentration of BC at surface. (Ramachandran and Rajesh, 2007; Kozlov et al., 2011; Chen et al., 2014). There is the absence of morning BC maximum in Moscow during rush hours, which was observed in many other cities, for example, in Tomsk (Kozlov et al., 2011), in Ahmedabad (Ramachandran and Rajesh, 2007), as well as in Athens (Diapouli et al., 2017). It can be explained by the specific regulation of diesel heavy trucks, which have a permission of entry only at night in Moscow (Popovicheva et al., 2020a). In addition, it

is necessary to account for a specific location of the MSU MO at a distance from the direct sources of urban emissions (highways).

## 4. Discussion

The analysis of urban aerosol pollution was made for a large agglomeration of the Moscow megacity using COSMO-Ru-ART model estimations and the results from the intensive measurement campaigns in April-May of 2018 and 2019 for a wide range

of meteorological and atmospheric air pollution conditions. We showed that the columnar aerosol characteristics during the intensive spring campaigns on the whole are close to those during the warm period of the year. However, AODs are slightly smaller compared with the average values for these months over the 2001-2020 period, which is in the agreement with the observed negative AOD trend in Moscow megacity (Zhdanova et a., 2020, Chubarova et al., 2016). A reduction in fine mode AOD fraction may be associated with a decrease in the emissions of urban aerosol precursor gases in recent years (Zhdanova

et al., 2020). A weak aerosol absorption in Moscow with relatively high values of SSA=0.94 corresponds to relatively small concentrations of black carbon (for urban areas) and its low $BC/PM_{10}$ ratio. The BC mass concentration is consistent with the estimated BC values in the GADS database for the Moscow region of 1.1 $\mu g\,m^{-3}$ during warm period (Koepke et al., 1997). This is twice as high compared with the BC concentrations in clean unpolluted regions (Herich et al., 2011), and more than 5 times smaller, than in the polluted areas of India and China (Singh et al. 2015; Wu et al., 2013). The concentrations of pollutants

typical for Moscow at surface layer are characterized by the reduced concentrations of sulfur oxides and by the increased concentrations of nitrogen oxides due to the emissions from transport and power plants (Report.., 2019, Report.., 2020). Mean $PM_{10}$ concentrations in Moscow correspond to those in large European cities (about 0.015-0.030 $mg\,m^{-3}$) and are significantly smaller than those in Asian industrial centers (Climate of Moscow.., 2017).

The analysis showed a noticeable day-to-day variability of different gas-and aerosol concentrations and columnar aerosol

characteristics. In some cases, especially high concentrations were observed at IPD=1, but in some cases smoke advection even in conditions with IPD=3 may provide elevated levels of aerosol pollution.

The correlation analysis between the parameters showed a statistically significant relationship between fine mode AOD500, which is more typical for urban aerosol (Dubovik et al., 2002), and surface mass concentrations of $PM_{10}$ and BC. At the same time, we observed a close relationship of columnar AOD and surface $PM_{10}$ with aerosol gas precursors. This may also indicate

the importance of secondary aerosol generation in the urban atmosphere of Moscow.

The atmospheric water vapor content W, which can be used as the indicator of the air mass (Myachkova, 1983), had close relationship with the aerosol parameters in the atmospheric column, which indicates more favorable processes of aerosol



formation in relatively warmer and wetter air masses (Chubarova, 2009). Wind speed had a statistically significant correlation with almost all surface and columnar aerosol characteristic, as well as aerosol gas precursor species. This could be explained
by the effects of ventilation of the urban environment, its better diluting and moving of the urban polluted air to the suburbs. The closer relationship between wind speed and BC compared to $PM_{10}$ indicates more important role of local meteorological conditions for black carbon. This happens, since BC is characterized by urban origin, while for $PM_{10}$ the regional background aerosol variations play an important role. The obtained negative relationship of BC and $BC/PM_{10}$ with wind speed is especially important, since it can serve an indicator of changes in the absorbing properties of the atmosphere. We also found the influence
of IPD on mean $BC/PM_{10}$ value, which is 0.7% at IPD=3, 5.5% at IPD=2, and 5.9% at IPD=1. The small $BC/PM10$ ratio at IPD=3 can be explained by the intensive air advection from the clean areas outside Moscow with low BC concentrations, and by strong vertical mixing with the upper layers of the atmosphere with smaller BC concentration. On contrary, at IPD=1, in conditions of stable stratified atmosphere and the absence of ventilation, we observed a strong increase in $BC/PM_{10}$. Thus, the use of the IPD index may significantly specify the $BC/PM_{10}$ ratio and, as a result, the absorbing properties of the atmosphere.
The analysis of the relationship between columnar AOD and surface $PM_{10}$ concentrations showed their correlation with a point of bifurcation around 0.05 $mg\,m^{-3}$, revealing the two types of relationships, that has physical explanation. The lower dependence characterizes the growth of $PM_{10}$ only in the close to surface atmospheric layer (due to local emissions) with predominant IPD=1 in the absence of the pronounced AOD increase. The upper dependence is associated with the influence of air mass advection, when the concentration of surface particles increases simultaneously with AOD. In (Gubanova et al.,
2018) close links between AOD and $PM_{2.5}$ concentrations were also obtained, especially during warm period on monthly scale. We also showed a significant increase in fine mode AOD fraction with $PM_{10,}$ and at $PM_{10}$>0.08 $mg\,m^{-3}$ fine mode AOD fraction is always higher 70%.

In the analysis, we paid more attention to BC concentrations, since this is the aerosol component, which significantly absorbs visible solar irradiance (Jacobson, 2004, 2006, Ramanathan and Carmichael, 2008). Based on the measured data, we obtained
simple regression equations to quantify BC concentration using the observed $PM_{10}$ or $NO_2$ mass concentrations for warm period conditions. These relationships were also confirmed by the model simulations.

We show that there is an inverse dependence of columnar aerosol single scattering albedo on the $BC/PM_{10}$ ratio according to both model and measurements but only in situation of the well-mixed atmosphere. Model estimates provide much higher correlation between these parameters and stronger sensitivity of SSA to the $BC/PM_{10}$ ratio. These dependences should be
studied further using larger statistics, since it can be important for indirect evaluation of absorbing aerosol properties in cloudy conditions, when the observations of SSA are not available from AERONET.

Using the results from parallel simultaneous AERONET measurements over the 2006-2020 period in Moscow and at the background Zvenigorod site we estimated the columnar urban aerosol component, $AOD_{urb}$ at 500 nm, which is about 0.025, and mainly (over 85%) consists of fine mode fraction (fine mode $AOD_{urb}$=0.021). The $AOD_{urb}$ estimates are in agreement with
our results obtained according to the first years of the urban aerosol studies of Moscow (Chubarova et al., 2011). Note, that this is the only one pair of collocated AERONET measurements in and out large megacity with long-term parallel measurements, operated by the same type of calibrated solar photometers, allowing a user to evaluate the columnar urban aerosol effects. The estimates of the urban AOD of about 0.02 were also made using hand-held sun photometer in Warsaw (Zavadzka et al., 2013). Warsaw with population of less than 2 million of people is much smaller than Moscow megacity.
However, the active use of coal with large $SO_2$ emission may provide significant increase in sulphate aerosol generation and, hence, lead to the increase in $AOD_{urb}$. For Moscow conditions the results of satellite MODIS estimates using the MAIAC multi-angle algorithm with accounting for surface reflectivity (Lyapustin et al., 2018) showed the urban AOD550 effect of about 0.01 (Zhdanova et al., 2020). Contrary, in (Li et al., 2018), according to the satellite MODIS data (MYD04_3 product) over Berlin much higher urban AOD (about 0.08) was obtained. We assume, that these $AOD_{urb}$ are likely to be overestimated,





since Berlin with a population of 3.6 million people and restriction on emissions should have lower aerosol pollution compared to Moscow megacity.

According to our estimates obtained during the intensive campaigns the urban AOD component at 550 nm was about 0.019 according to measurements, and 0.015 according to modelling results. These values are consistent in general with our long-term estimates shown above, as well as with the estimates in (Chubarova et al., 2011). The urban AOD fraction

$AOD_{urb}/AOD_{meas}$ is equal to 19% according to the measurements, and to 15% - according to the calculations. However, in conditions with relatively low AOD, the urban AOD fraction may exceed 50%.

The model $PM_{10urb}$ provides some underestimation compared with measurements (respectively, 0.006 $mgm^{-3}$ and 0.016 $mgm^{-3}$). On the contrary, there was a model overestimation of BC concentration (1.6 $\mu g m^{-3}$ compared with 0.95 $\mu g\ m^{-3}$), which may be the cause of the observed too low values of model single-scattering albedo. This difference of model estimates with the

observations is occurred likely due to underestimating of primary aerosol emissions and overestimating of BC emissions in the TNO2010 inventory for Moscow megacity, which should be studied further.

We also obtained the increase of the urban aerosol component with the growth of the total AOD or $PM_{10}$, which indicates the presence of favorable conditions for secondary aerosol generating for both natural and urban components. This is also in accordance with a statistically significant correlation of AOD and $PM_{10}$ with the aerosol gas precursors. At the same time,

closer dependence is observed between the measured $PM_{10urb}$ and total $PM_{10}$ at surface layer, where the emissions of polluted substances are taken place. We also have a noticeable relationship between the calculated and measured $PM_{10urb}$, which significantly depend on the IPD index.

The increase in aerosol loading in the background clean conditions at the ZSS in Zvenigorod can also occur due to the advection of polluted air from Moscow. After the removal of the cases affected by Moscow pollution the number of negative $AOD_{urb}$

became much smaller and they did not exceed 0.01, which is close to the uncertainty of sun photometer measurements. Thus, for Moscow conditions according to the measurements the average aerosol urban pollution comprises the estimates for $AOD_{urb}$=0.019, for $PM_{10urb}$=0.016 $mgm^{-3}$, and for BC =0.95 $\mu g m^{-3}$.

The analysis of diurnal cycle for $PM_{10urb}$ and $BC_{urb}$ at surface layer have detected their noticeable changes in both model and experimental data. We revealed the significant accumulation of $PM_{10}$ and BC at night below the inversion layer in conditions

with IPD=1. In these conditions at night the increase in concentration reached 4 times for $PM_{10urb}$ (30-40 $\mu g m^{-3}$), and 3 times (up to 3 - 3. 5 $\mu g m^{-3}$) - for BC compared with conditions at IPD=3. During warm period there is a noticeable increase in the height of boundary layer during daytime, which contributes to the processes of dilution and amplification of convection, when the earth's surface is heated by solar radiation (Ramachandran and Rajesh, 2007; Kozlov et al., 2011; Chen et al., 2014) providing a decrease in surface concentrations of different aerosol and gas species. The observed $AOD_{urb}$ does not have a

clear daily cycle, however, according to model calculations, it is slightly higher at night, especially in conditions of a stratified atmosphere at IPD=1.

## Conclusions

We have presented a detailed analysis of surface and columnar aerosol measurements and model simulations in urban and clean background conditions, which allows us to obtain reliable quantitative estimates of the urban component of aerosol

pollution at surface and in the atmospheric column and to identify the relationships between them in different meteorological conditions.

The correlation analysis between the parameters showed a statistically significant relationship between fine AOD500 mode and surface mass concentrations of $PM_{10}$ and BC as well as with aerosol gas precursors. Both model and experimental datasets



have shown a statistically significant linear correlation of BC with $NO_2$ and $PM_{10}$ mass concentrations, which indicates mostly common sources of emissions of these substances.

The average urban component of AOD ($AOD_{urb}$) at 500nm for Moscow obtained over the 2006-2020 period of parallel measurements in Moscow and background Zvenigorod conditions is about 0.025 with more than 85% of fine mode fraction. According to the measurements we obtained the mean estimates of $AOD_{urb}$=0.019, $PM_{10urb}$=0.016 mg m$^{-3}$, and BC =0.95 $\mu$g m$^{-3}$. A similar level of $AOD_{urb}$ =0.015 was evaluated using model simulations. We showed that $AOD_{urb}$ in Moscow comprised about 15-19% of total AOD at 550nm, but in some cases, it may exceed 50%.

There was a pronounced increase in the BC/$PM_{10}$ ratio from 0.7% to 5.9% with the decrease in IPD index related to the amplification of the atmospheric stratification. We also found an inverse dependence between the BC/$PM_{10}$ ratio and columnar single scattering albedo (SSA) for the intense air mixing conditions. This dependence together with the obtained negative correlation between wind speed and BC/$PM_{10}$ may serve an indicator of changes in the absorbing properties of the atmosphere due to meteorological factors.

A pronounced diurnal cycle of $PM_{10urb}$ and urban BC, and their strong correlation with the intensity of particle dispersion indices have been obtained. At night a significant accumulation of $PM_{10}$ and BC below the inversion layer is observed in conditions with IPD=1, reaching 4 times for $PM_{10urb}$, and 3 times - for BC compared with conditions at IPD=3. The observed $AOD_{urb}$ does not have a clear daily course, however, according to model calculations, it is slightly higher at night, especially in poorly mixed conditions at IPD=1.

In future work, we plan to use the obtained results for evaluating the radiative effects of the urban aerosol pollution and for identifying its influence on meteorological parameters and weather forecast.

**Author contribution**: The conceptualization, data analysis, and final text writing was fulfilled by N. E. Chubarova. A.A. Kirsanov, G.S. Rivin, B.Vogel, H. Vogel designed the model experiments, performed the simulations, O.B. Popovicheva contributed with the BC dataset, E.E. Androsova contributed with data analysis, and the design of Figures. N.E. Chubarova prepared the manuscript with contributions from all co-authors.

**Competing interests**: The authors declare that they have no conflict of interest.

**Acknowledgements.** We are grateful for the support of the Government of the Russian Federation under the grant No. 075-15-2021-574. This research was performed according to the Development program of the Interdisciplinary Scientific and Educational School of Lomonosov Moscow State University « Future Planet and Global Environmental Change» and was carried out using the equipment of MSU Shared Research Equipment Center "Technologies for obtaining new nanostructured materials and their complex study" and purchased by MSU in the frame of the Equipment Renovation Program (National Project "Science").

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






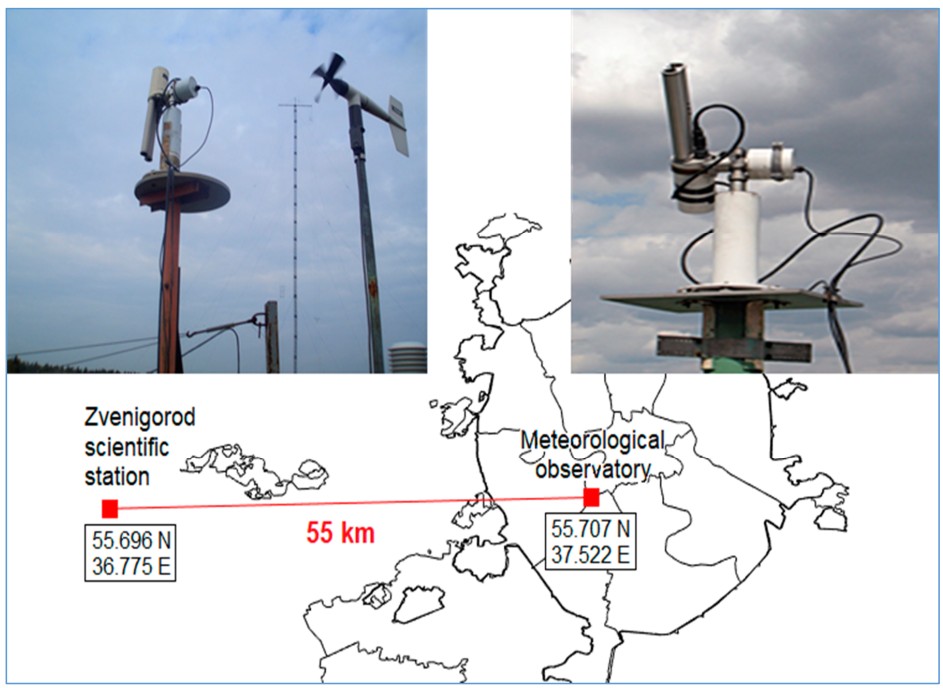

**Figure 1: Location of Cimel sun/sky AERONET photometers at the Meteorological Observatory of Moscow State University (MSU MO) and at the Zvenigorod Scientific Station (ZSS) of the A.M. Obukhov Institute of Atmospheric Physics. Moscow region.**




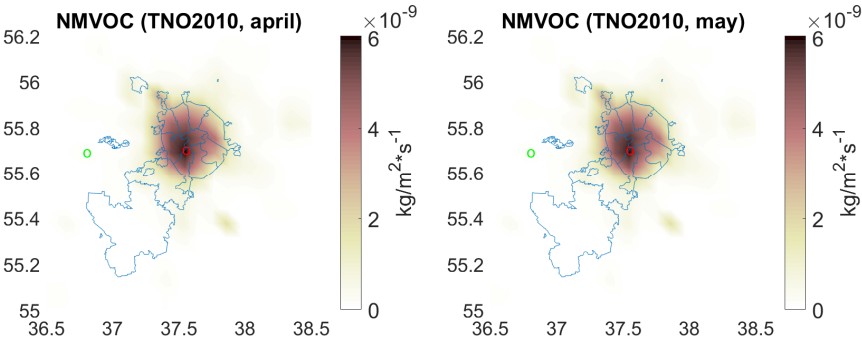


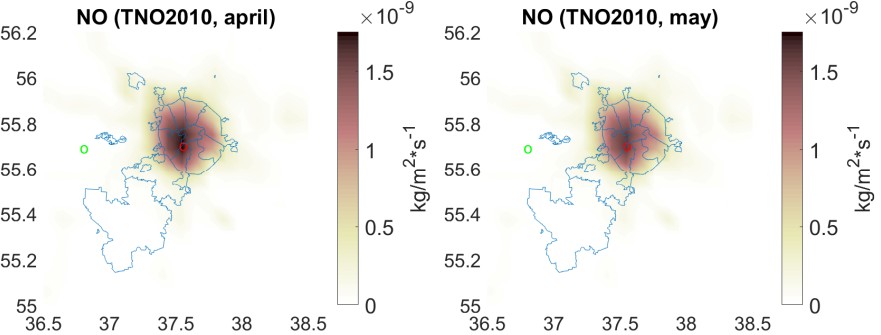




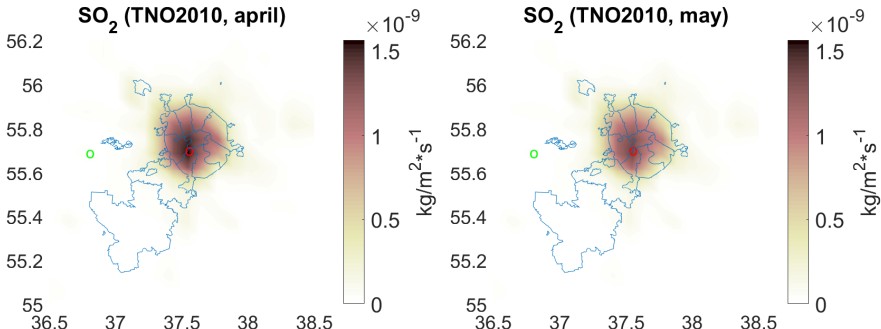

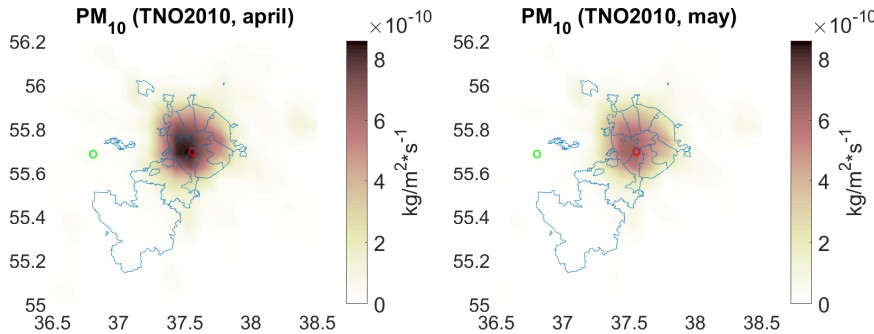

**Figure 2: Monthly mean emissions of aerosol gas-precursors according to the TNO2010 inventory in April and May in Moscow region.**





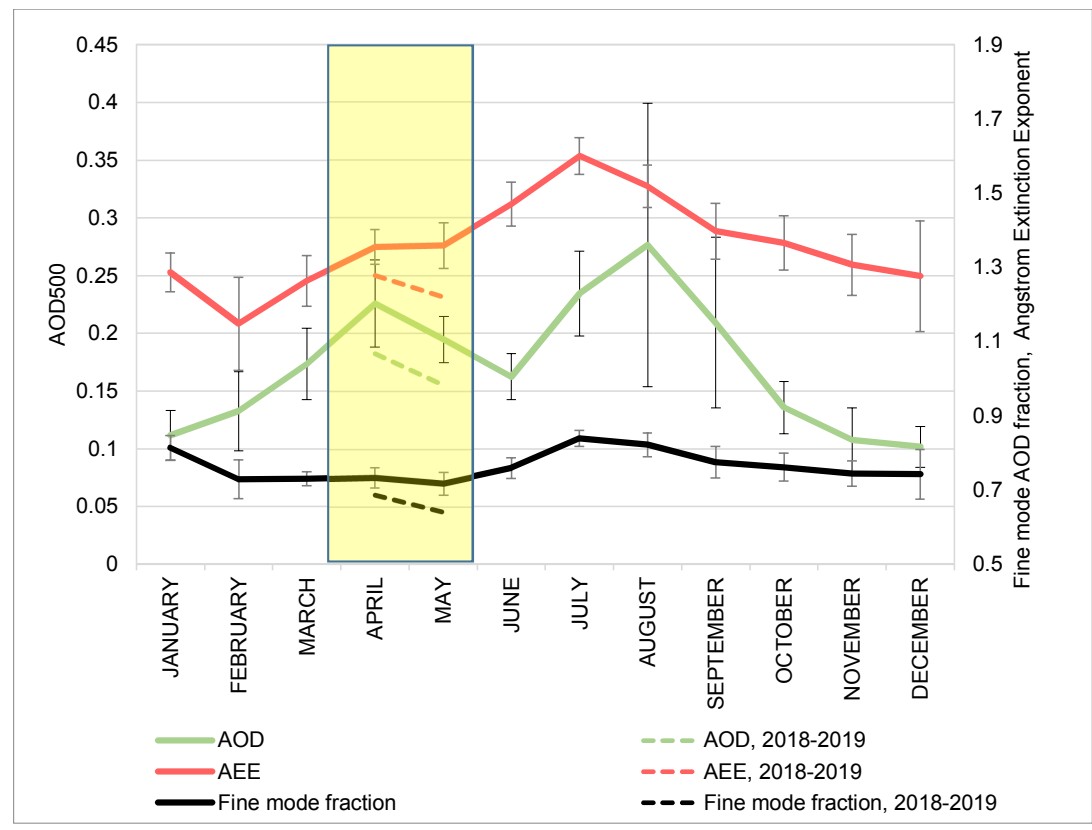

**Figure 3: Seasonal changes in monthly mean AOD at 500 nm, Angstrom Extinction Exponent (AEE) at 440-870 nm interval, fine mode AOD fraction at 500 nm for the 2001-2020 period and for April-May months in 2018 and 2019. Level 2, version 3. Moscow, MSU MO. The period of the study is shown by yellow column.**





(a)

(b)

**Figure 4: Daily means of AOD500, fine mode AOD fraction at 500 nm, PM₁₀ and aerosol gas precursors (NOx, CHx)**
**mass concentrations (in mg m⁻³), BC/PM₁₀ ratio, water vapor content (in cm) and IPD index in 2018 (a) and 2019 (b).**





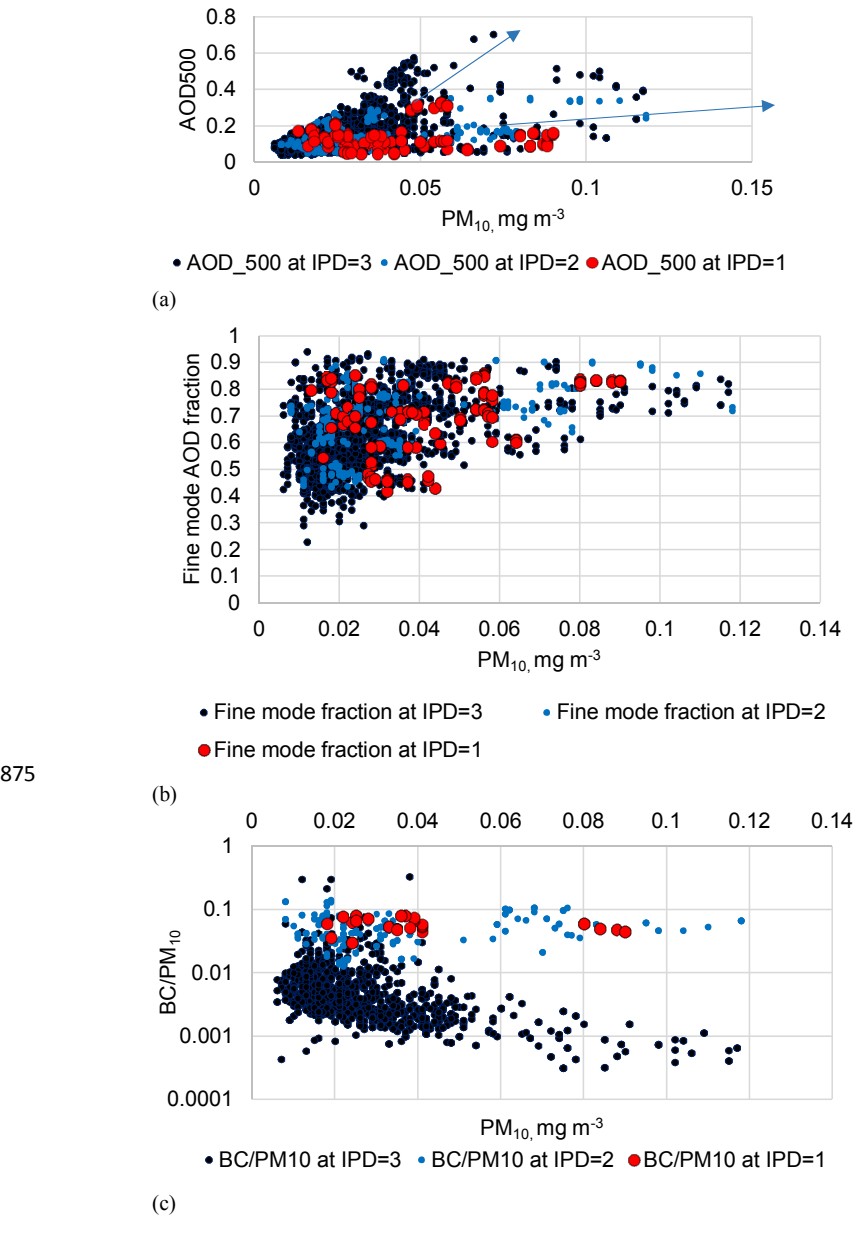

Figure 5: AOD at 500 nm (a), fine mode fraction of AOD at 500nm (b), and BC/PM$_{10}$ ratio (c) as a function of PM$_{10}$ mass concentration (mg m$^{-3}$) under various IPD mixing air conditions.






Figure 6: The dependence of measured (left column) and model (right column) BC mass concentration as a function of $PM_{10}$ (a,d), and aerosol gas precursors ($NO_2$ - b, e; $SO_2$– c, f) for different *IPD* regimes for April-May 2018. The cases affected by biomass burning aerosol were excluded.



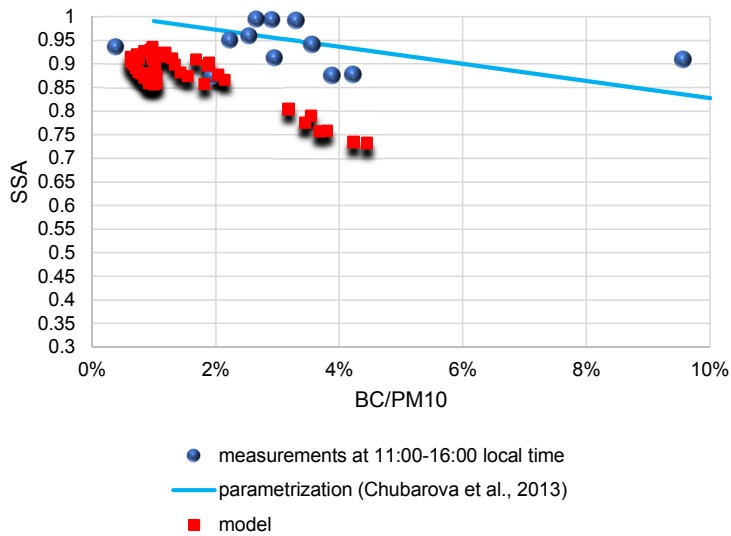

**Figure 7: Single scattering albedo in visible spectral region as a function of BC/PM$_{10}$ ratio according to model simulations and measurements within 3 hours around local noon, and linear regression obtained from observations in (Chubarova et al., 2013). Clear sky conditions.**






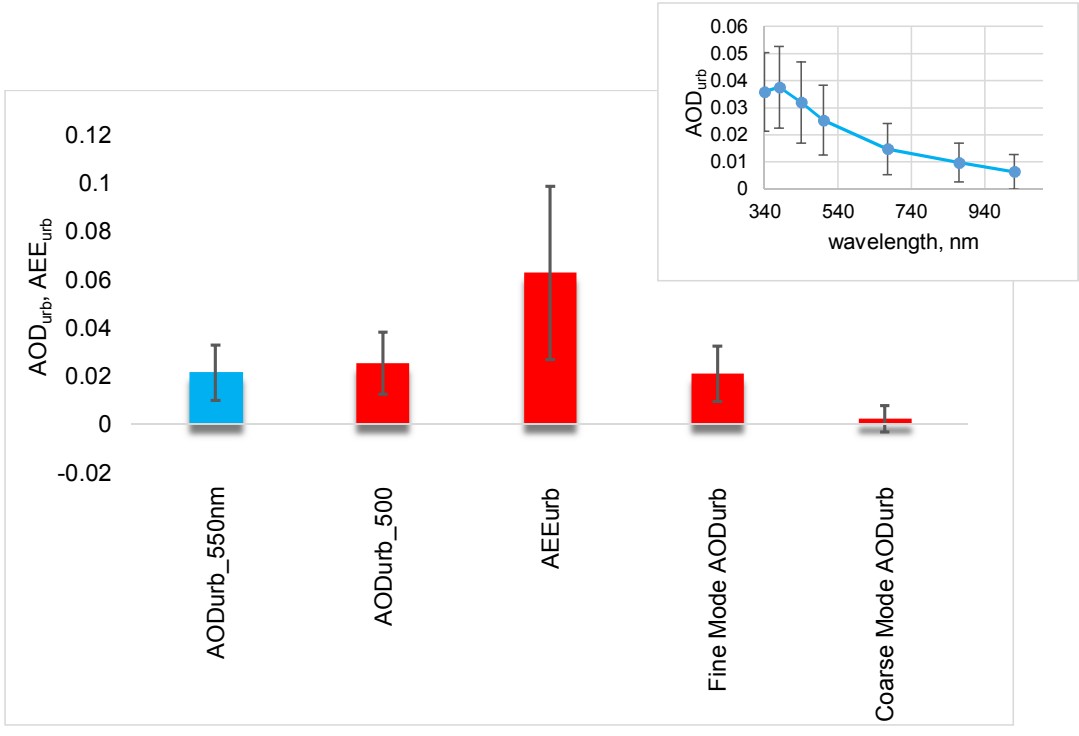

**Figure 8: Monthly mean urban components of different aerosol parameters - AOD$_{urb}$ at 550nm and 500 nm, Fine Mode**
**AOD$_{urb}$, Coarse Mode AOD$_{urb}$, urban component of the Angstrom Extinction Exponent AEE$_{urb}$ with confident intervals at 0.05 significance level. The inset shows the mean AOD$_{urb}$ spectral dependence. Comment: we show the AOD at two wavelengths to provide more convenient comparisons with the CIMEL sun-photometer observations (AOD at 500 nm) and model results (AOD at 550 nm). Moscow. 2006-2020.**



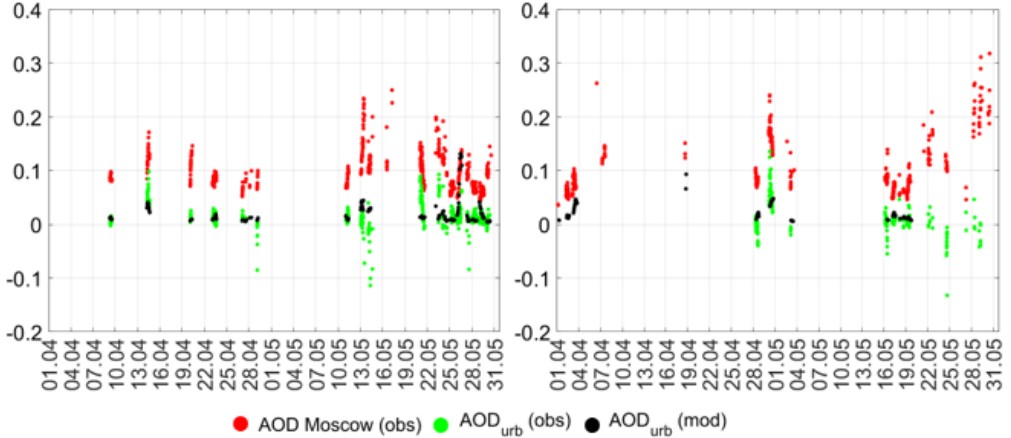

(a).

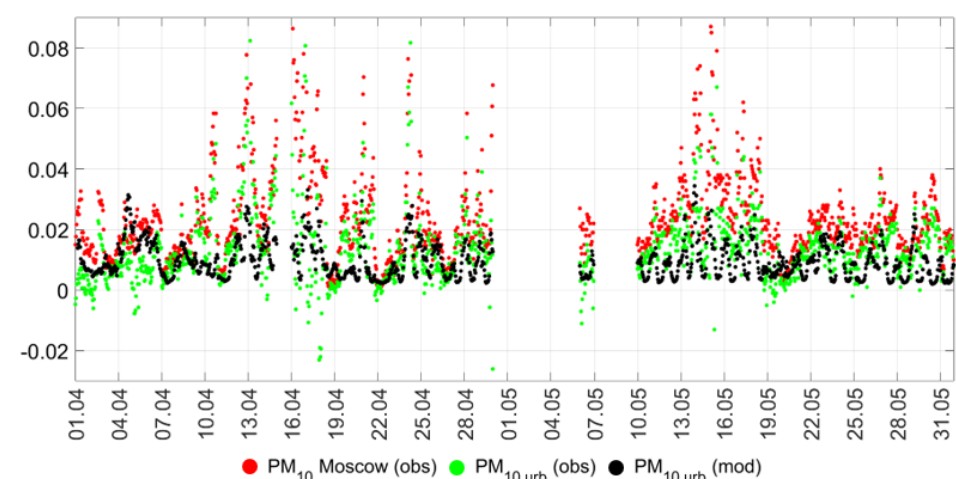

(b).

915        **Figure 9: (a) - Time series of AOD at 550 nm simulated using direct observations of AOD at 500nm and AEE at 440-870nm, and the AOD$_{urb}$ component according to measurements and modelling in 2018 (left upper panel) and 2019 (right upper panel); (b) – Time series of PM$_{10}$ (in mg m$^{-3}$) in Moscow and urban component of PM$_{10}$ according to measurements, PM$_{10urb}$(obs), and modelling, PM$_{10urb}$(mod) (in mg m$^{-3}$). 2018. We used a filter of total cloud amount N<5. The cases affected by biomass burning aerosol were excluded.**






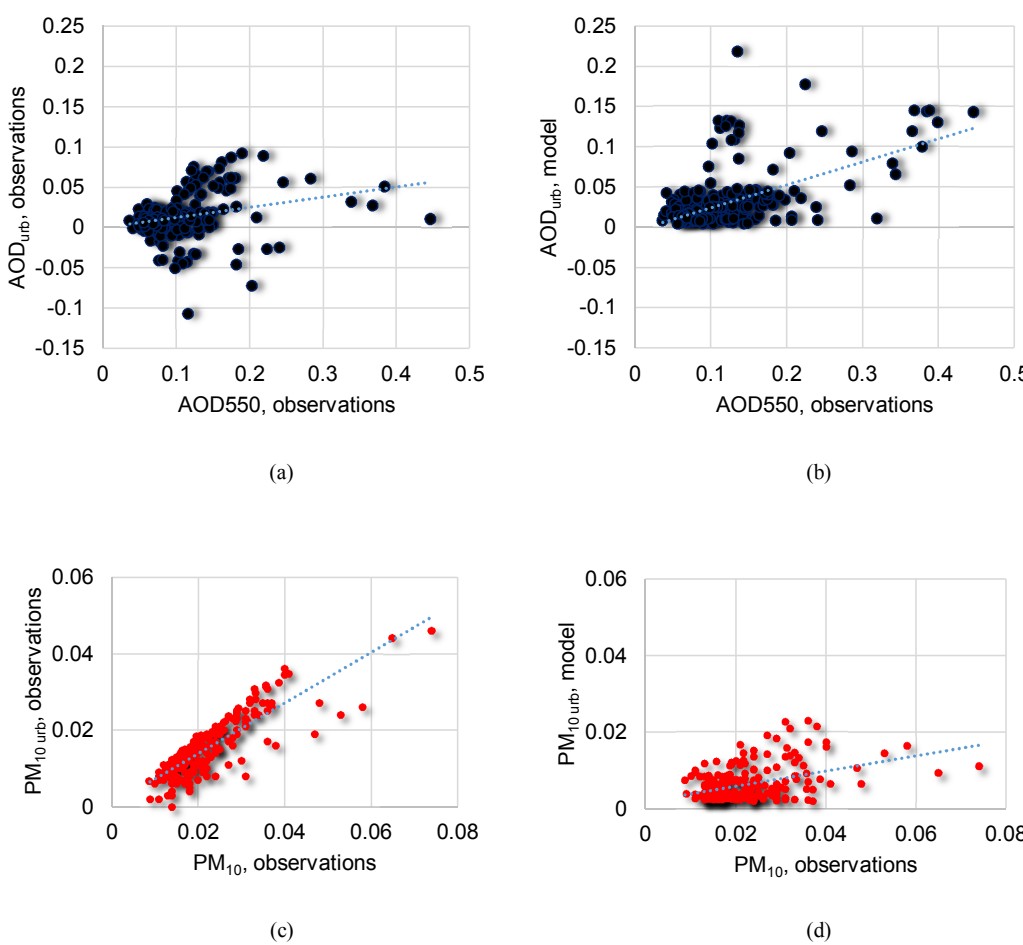

Figure 10: Measured (a,c) and model (b,d) urban component of aerosol optical depth at 550 nm ($AOD_{urb}$) and urban $PM_{10}$ mass concentration ($PM_{10urb}$, in $mg\,m^{-3}$) as a function of the observed total AOD at 550 nm (a,b) and $PM_{10}$ (c,d) in Moscow (MSU MO). For consistency reason we show only quasi-simultaneous AOD and $PM_{10}$ measurements during the daytime period with AOD observations. The cases affected by biomass burning aerosol were excluded. Clear sky conditions.






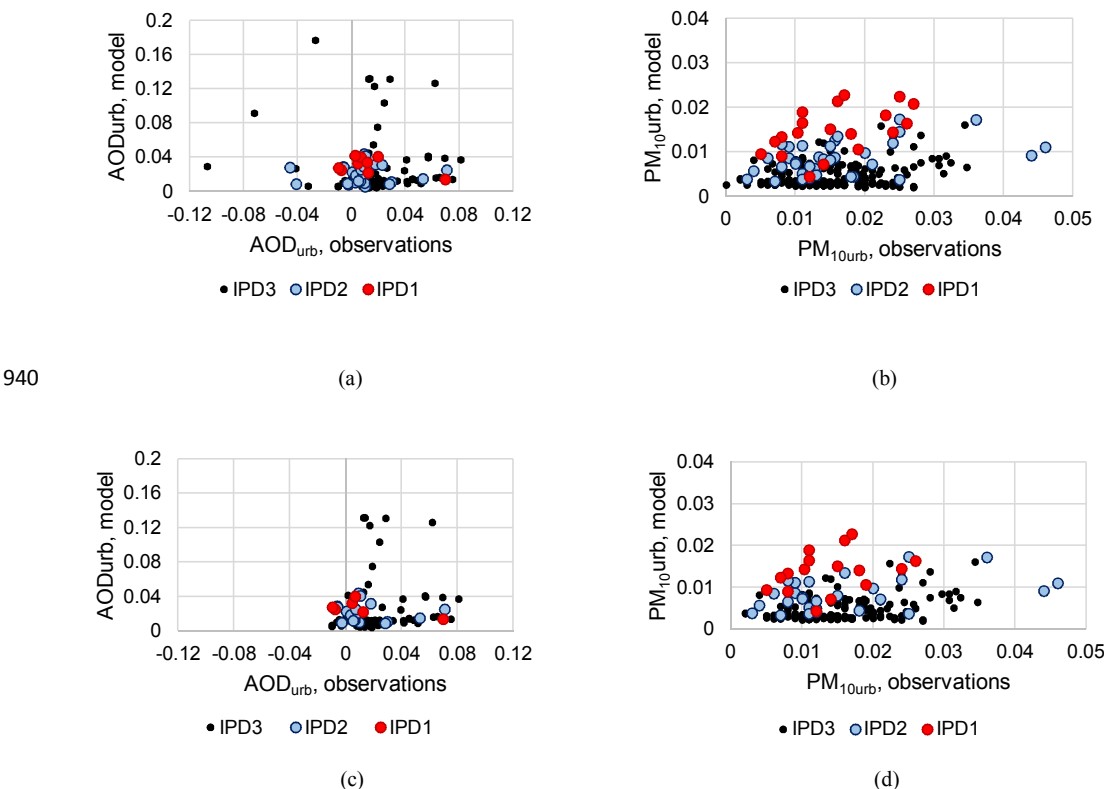

940                                                        (a)                                                                          (b)

(c)                                                                          (d)

**Figure 11: The relationship between model and measured urban aerosol optical depth at 550 nm (AOD_urb - a,c) and urban component of $PM_{10}$ ($PM_{10urb}$, in mg m$^{-3}$ - b,d) for all cases (n=229) (a,b) and for the cases without the effects of**

**urban air advection from Moscow to Zvenigorod (n=203) (c,d) at different IPD conditions.  For consistency reason we show only quasi-simultaneous AOD and $PM_{10}$ measurements during the daytime period with AOD observations. The cases affected by biomass burning aerosol were excluded. 2018.**



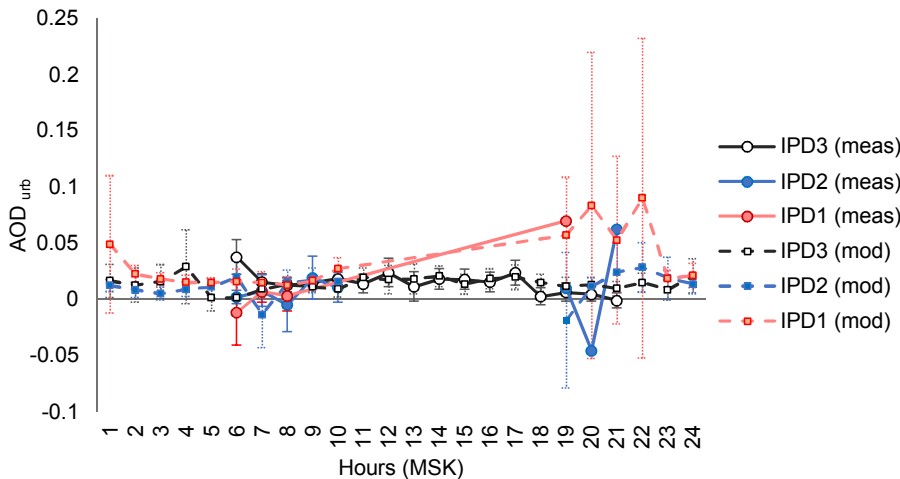


(a)

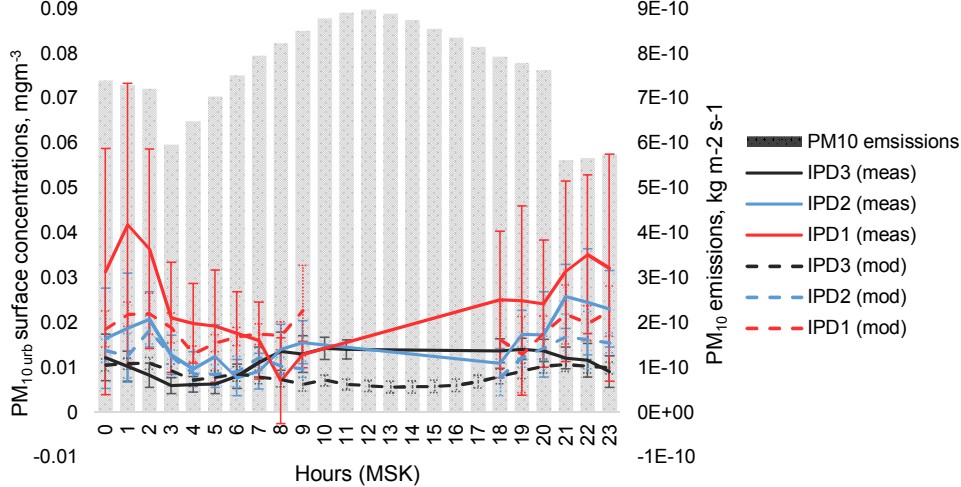

(b)






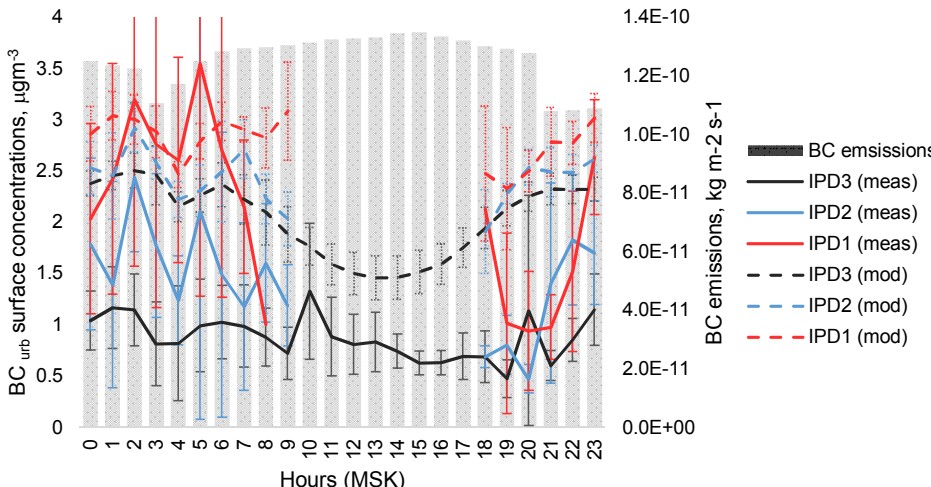

(c)

**Figure 12: Diurnal cycle of AOD $_{urb}$(a), PM$_{10\,urb}$ (b) and BC urb mass concentrations (c) according to measurements and modelling for different conditions of the intensity of particle dispersion (IPD). For PM$_{10}$ and BC the diurnal cycle of their emissions is also shown. The confidence intervals were calculated at 0.05 significance level. Moscow.**



**Table 1.** Statistics of hourly mean aerosol characteristics in the total column of the atmosphere including aerosol optical depth (AOD at 500nm), Angstrom Extinction Exponent, Fine mode fraction at 500 nm, single scattering albedo (SSA) at 675nm, factor of asymmetry (ASY) for fine, coarse and total aerosol at 675nm, water vapor content (W, cm) and mass concentrations of $PM_{10}$, BC, different aerosol gas-precursors, and $BC/PM_{10}$ ratio. Moscow. April-May, 2018-2019.

| | Mean value | Median | Confident interval at 0.05 | Minimum | Maximum | Total case number |
|---|---|---|---|---|---|---|
| AOD at 500 nm | 0.17 | 0.12 | 0.01 | 0.04 | 1.00 | 736 |
| Angstrom extinction exponent, AEE | 1.23 | 1.22 | 0.02 | 0.27 | 1.91 | 736 |
| Fine mode AOD500 fraction | 0.65 | 0.64 | 0.01 | 0.24 | 0.94 | 710 |
| SSA675 | 0.93 | 0.94 | 0.01 | 0.80 | 0.99 | 48 |
| ASY675_fine | 0.53 | 0.53 | 0.01 | 0.48 | 0.60 | 52 |
| ASY675_coarse | 0.81 | 0.81 | 0.01 | 0.72 | 0.91 | 52 |
| ASY675_total | 0.64 | 0.63 | 0.01 | 0.57 | 0.71 | 52 |
| W, cm | 1.07 | 1.08 | 0.04 | 0.35 | 2.65 | 580 |
| $PM_{10}$, $mg\,m^{-3}$ | 0.028 | 0.025 | 0.001 | 0.000 | 0.174 | 2892 |
| BC, $\mu g\,m^{-3}$ | 1.36 | 1.03 | 0.05 | 0.004 | 8.894 | 2054 |
| $BC/PM_{10}$*100, % | 4.7 | 4.3 | 0.116 | 0.024 | 26.6 | 2019 |
| CO, $mg\,m^{-3}$ | 0.229 | 0.193 | 0.006 | 0.000 | 1.273 | 2869 |
| $SO_2$, $mg\,m^{-3}$ | 0.002 | 0.002 | 0.000 | 0.000 | 0.038 | 2466 |
| CHx, $mg\,m^{-3}$ | 1.473 | 1.450 | 0.004 | 1.310 | 2.970 | 2852 |
| NOx, $mg\,m^{-3}$ | 0.041 | 0.030 | 0.001 | 0.000 | 0.330 | 2902 |
| NO, $mg\,m^{-3}$ | 0.006 | 0.001 | 0.001 | 0.000 | 0.211 | 2902 |
| $NO_2$, $mg\,m^{-3}$ | 0.036 | 0.028 | 0.001 | 0.000 | 0.154 | 2902 |
| $O_3$, $mg\,m^{-3}$ | 0.072 | 0.072 | 0.001 | 0.000 | 0.176 | 2902 |

Note: the case number is different, since columnar aerosol characteristics can be measured only during daytime and some of

them – only in semi-clear sky conditions.





**Table 2.** Correlation matrix between hourly mean different aerosol characteristics, aerosol gas- precursors, and meteorological parameters. April-May, 2018-2019. N=230. Statistically significant correlation coefficients at a significance level of 0.05 is shown in bold.

| | AOD500 | Fine AOD500 mode | Coarse AOD500 mode | IPD | Wind Speed | BC | PM$_{10}$ | BC/PM$_{10}$ | W, cm | Angstrom extinction exponent, AEE | CHx | CO | NO | NO$_2$ | SO$_2$ |
|---|---|---|---|---|---|---|---|---|---|---|---|---|---|---|---|
| AOD500 | 1.00 | **0.98** | **0.57** | 0.03 | **-0.21** | **0.34** | **0.57** | -0.08 | **0.21** | **0.58** | **0.34** | **0.27** | **0.20** | **0.29** | **0.17** |
| Fine AOD500 mode | | 1.00 | **0.39** | 0.01 | **-0.27** | **0.39** | **0.58** | -0.06 | **0.13** | **0.70** | **0.40** | **0.25** | **0.24** | **0.33** | **0.13** |
| Coarse AOD500 mode | | | 1.00 | 0.12 | 0.12 | -0.01 | **0.23** | **-0.14** | **0.42** | **-0.18** | -0.03 | **0.22** | -0.04 | 0.00 | **0.26** |
| IPD | | | | 1.00 | **0.48** | **-0.24** | -0.05 | -0.10 | **-0.21** | **-0.17** | **-0.19** | **-0.15** | -0.09 | **-0.19** | **0.13** |
| Wind Speed | | | | | 1.00 | **-0.49** | **-0.25** | **-0.26** | -0.05 | **-0.43** | **-0.44** | **-0.22** | **-0.34** | **-0.47** | 0.04 |
| BC | | | | | | 1.00 | **0.64** | **0.58** | -0.04 | **0.44** | **0.70** | **0.39** | **0.70** | **0.70** | 0.04 |
| PM$_{10}$ | | | | | | | 1.00 | -0.11 | -0.09 | **0.42** | **0.75** | **0.32** | **0.66** | **0.70** | 0.11 |
| BC/PM$_{10}$ | | | | | | | | 1.00 | 0.04 | 0.06 | 0.06 | 0.08 | 0.10 | 0.10 | -0.04 |
| W, cm | | | | | | | | | 1.00 | 0.12 | **-0.13** | **0.16** | **-0.19** | **-0.21** | 0.04 |
| Angstrom extinction exponent, AEE | | | | | | | | | | 1.00 | **0.46** | **0.24** | **0.26** | **0.33** | -0.08 |
| CHx | | | | | | | | | | | 1.00 | **0.41** | **0.77** | **0.77** | 0.07 |
| CO | | | | | | | | | | | | 1.00 | **0.39** | **0.35** | 0.06 |
| NO | | | | | | | | | | | | | 1.00 | **0.87** | **0.24** |
| NO$_2$ | | | | | | | | | | | | | | 1.00 | **0.33** |
| SO$_2$ | | | | | | | | | | | | | | | 1.00 |




**Table 3**. Main statistics of hourly average aerosol characteristics and their urban components after removing the cases of smoke advection, and the effects of urban air advection from Moscow.

| | Average | Q2 (50% quantile) | Q1 (25% quantile) | Q3 (75% quantile) | Minimum value excluding outliers | Maximum value excluding outliers | Case number |
|---|---|---|---|---|---|---|---|
| AOD at 550nm measurements, Moscow, 2018-2019 | 0.098 | 0.090 | 0.070 | 0.118 | 0.015 | 0.192 | 168 |
| $AOD_{urb}$ measurements 2018-2019 | 0.019 | 0.013 | 0.005 | 0.023 | -0.023 | 0.051 | 168 |
| $AOD_{urb}$ model 2018-2019 | 0.015 | 0.008 | 0.006 | 0.015 | 0.002 | 0.027 | 168 |
| BC measurements, Moscow $\mu g m^{-3}$ 2018-2019 | 0.946 | 0.698 | 0.414 | 1.135 | 0.007 | 2.191 | 129 |
| BC model, $\mu g/m^3$ 2018-2019 | 1.590 | 1.375 | 1.064 | 1.996 | 0.911 | 3.201 | 129 |
| $BC_{urb}$ model, $\mu g m^{-3}$ 2018-2019 | 1.465 | 1.288 | 1.018 | 1.798 | 0.862 | 2.920 | 129 |
| $PM_{10}$ measurements, Moscow $mg m^{-3}$ 2018 | 0.023 | 0.019 | 0.016 | 0.029 | 0.007 | 0.050 | 163 |
| $PM_{10urb}$ measurements, $mg m^{-3}$ 2018 | 0.0159 | 0.014 | 0.011 | 0.020 | 0.002 | 0.032 | 163 |
| $PM_{10urb}$ model, $mg m^{-3}$ 2018 | 0.006 | 0.005 | 0.003 | 0.008 | 0.002 | 0.014 | 163 |
| $PM_{10}$ model 2018 | 0.007 | 0.005 | 0.003 | 0.009 | 0.002 | 0.017 | 163 |
| *$AOD_{urb}$ measurements (all cases with urban air advection) 2018-2019* | *0.016* | *0.012* | *0.006* | *0.023* | *-0.018* | *0.041* | *200* |
| *$PM_{10urb}$ measurements, $mg m^{-3}$ (all cases with urban air advection) 2018* | *0.0158* | *0.014* | *0.011* | *0.020* | *0* | *0.032* | *197* |

Note: we used a filter of total cloud amount N<5. We also consider only the cases with the AOD data during daytime for evaluating the $PM_{10}$ statistics and no effects from biomass burning aerosol.
