# Peer review of "Columnar and surface urban aerosol in Moscow megacity according to measurements and simulations with COSMO-ART model"

_Atmospheric Chemistry and Physics, 2022_

## Author Comment (AC1)

We would like to thank the reviewer for the very useful comments, which help to improve the quality of the text. By green color we show edited or new text in the updated variant.

Anonymous Referee #1
Referee comment on "Columnar and surface urban aerosol in Moscow megacity according to measurements and simulations with COSMO-ART model" by Natalia Chubarova et al., Atmos. Chem. Phys. Discuss., https://doi.org/10.5194/acp-2022-83-RC1, 2022
The paper assesses the Moscow megacity's aerosol pollution, focusing on urban air pollution and its radiative effect. The authors employed both experimental and modeling approaches to give a comprehensive picture of the magnitude and temporal variability of urban pollution in Moscow. The paper is well written and logically organized. It has a comprehensive introduction and discusses new valuable observation results. It could be published after a minor revision according to the below comments.

General comments:

*1/ The model itself and especially its aerosol component should be better explained. It would be helpful to discuss the spatial maps from the model to see how representative (spatially) the observations are.*

Thank you. Yes, we added additional information about the details of aerosol evaluation in the model.

*The new text is the following:*

"The simulations of reactive gaseous and particulate matter are based on the enhanced KAMM/DRAIS/MADEsoot/dust model (Riemer et al., 2003a; Vogel et al., 2006, Vogel et al., 2010). In MADEsoot (Modal Aerosol Dynamics Model for Europe extended by soot) all aerosol modes are represented by lognormal distributions. Five modes for Aitken and sub-micron aerosol particles include one pure soot mode, secondary particles consisting of sulphate, ammonium, nitrate, organic compounds (SOA), as well as the modes representing aged soot particles, consisting of sulphate, ammonium, nitrate, organic compounds, water, and soot. It also includes coarse particle mode, which contains additional anthropogenic emitted particles. All aerosol fractions are subject to coagulation and condensation following Binkowski and Shankar (1995), Whitby et al. (1991), Kerminen and Wexler (1994). Schell et al. (2001). Odum et al. (1996). The soot particles are directly emitted into the atmosphere. Coagulation and condensation are accounted for in transfer of soot from external into internal mixture. The Aitken and sub-micron particles are formed due to the aging process. For each mode prognostics equations for the number density and the mass concentration are solved numerically. The standard deviations are kept constant. Since the number densities of the coarse mode are small the inter-modal coagulation between the coarse mode and the other modes and the intra-modal coagulation of the coarse mode particles are both neglected. Additionally, the aerosol distributions are modified by the sedimentation, advection and turbulent diffusion processes. More details can be found in (Vogel et al., 2010). The resuspension of urban dust with stronger winds currently is not taken into account. The chemical reactions of the gaseous species are calculated using the chemical mechanism RADMKA (Regional Acid Deposition Model Version Karlsruhe) based on RADM2 (Regional Acid Deposition Model; Stockwell et al., 1990) with the important updates described in (Vogel et al., 2010). The photolysis frequencies were simulated according to (Vogel et al., 2009). For

the evaluation of the aerosol optical properties (the extinction coefficient, the single scattering albedo and the asymmetry factor) a special parameterization scheme is used based on the a priori calculations with the application of the approach described in (Bohren and Huffmann, 1983) and pre-calculated aerosol distributions. This procedure is based on typical size distributions and chemical compositions, which are simulated in the model domain."

Concerning the spatial maps: we added location of urban and pristine sites in Figure 2, where the spatial fields of emissions are shown. This clearly proves that the locations of both site are really typical for urban ( MOscow MSU) and clear conditions ( Zvenigorod). Please, find the updated Fig.2 in the supplemental file.

[Figure]

[Figure]

[Figure]

**Figure 2: Monthly mean emissions of aerosol gas-precursors, PM10 and BC emissions according to the TNO2010 inventory in April and May in Moscow region. The location of the Moscow State University Meteorological Observatory is shown by red circle and Zvenigorod site – by green circle.**

*2/ I am not clear how does the model with 7-km grid spacing describe in-city aerosol concentration? Is any parameterization for the urban terrain used?*

Yes, you are right. Local in-city variations are not represented by the model with this resolution, but the difference between the city and suburb stations provide reliable estimates of urban effects. The main aim

of this study was to evaluate the features of air pollution using the difference between the data at Moscow State University Meteorological Observatory and at upwind background site at Zvenigorod (around 55 km to the west). Therefore, we do not analyze the spatial structure of pollution within the Moscow itself. In this situation it is possible to use 7 km grid. In this study no parameterization for the urban terrain is used but in our current studies with 2-km grid spacing it has been applied, along with TERRA_URB parameterization.

We added in the text the following clarification:

"Since the main objective of our study was to evaluate aerosol pollution in Moscow at the MSU MO and at upwind background conditions, we did not focus on detailed variations of aerosol inside the city and used 7 km grid step for model simulations."

*3/ PM2.5 was not measured, and all information about a fine aerosol fraction came from AERONET retrieval?*
*It is unclear what are the natural aerosols that contribute to PM10?*

Only the measurements of $PM_{10}$ mass concentrations are currently available. However, according to Wu and Boor (2021, included in reference list) fine aerosol mode is the predominant fraction of PM10 in urban aerosol in Central and Northern Europe (see, for example, see Fig. 10 in this paper). So, we can consider that PM10 is mainly consisted of fine aerosol mode in Moscow as well, similar to columnar aerosol. Unfortunately, no information is available concerning the measurements of the properties of natural aerosol in Moscow background, except its total mass concentrations at surface. However, if take, in the first approximation, the data of CAMS reanalysis (Bozzo et al., 2020) for Moscow background conditions, we obtain that organic aerosol during warm period is mainly (more than 50%) contributed to the total amount of natural aerosol there. Sulphate aerosol is the second component in the range (20-40%), and the dust contribution lies within 0-14%.

*4/ The model does not account for any natural emissions, including urban dust.*
*If observations are not available, it will help discuss aerosol vertical distributions, at least*
*from the model perspective.*
The model accounts for relatively small natural biogenic emissions of hydrocarbons in background region in spring (see Fig.2 for NMVOC emissions over the whole Moscow region), which are the gas precursors of organic aerosol. In addition, urban emissions include direct emission of particulate matter of undefined composition (as dust fraction) and soot, as well as the gas-aerosol precursors for sulphate, organic, and nitrate secondary aerosols. From the model perspective anthropogenic aerosol is emitted from the bottom levels of the model (up to 990 m) and decreases with height (see Figure below).

[Figure]

Figure 1C. This is an illustration of the PM10 vertical distribution (15/04/2018, 12UTC).

We thank you for the idea to study the vertical profiles of aerosol distribution, which we will certainly do in the separate paper. However, in this paper we focus mainly on the difference in aerosol characteristics between urban and background sites. That is why we do not account for the natural aerosol, which should be similar at these sites and will not affect the evaluation of urban aerosol component.

Some additional clarification has been added in the text:
"Urban aerosol sources according to TNO2010 include direct emission of particulate matter of undefined composition (as dust fraction) and soot, as well as the gas-aerosol precursors of sulphate, organic, and nitrate secondary aerosols. The model also accounts for relatively small natural biogenic emissions of non-methane volatile organic compounds from the Global Land Cover 2000 project, which are the gas precursors of organic aerosol."

Specific comments
*L10: COSMO-ART is a regional meteorology and chemical transport model*

Following (Vogel et al., 2009), where the first mentioning of the COSMO –ART was made, we specified the name of this model in the following way in this place and everywhere in the text:

"Urban aerosol pollution was analyzed over the Moscow megacity region using COSMO-ART (COSMO — COnsortium for Small-scale MOdelling, ART — Aerosols and Reactive Trace gases) online coupled mesoscale model system.."

*L39: The aerosol forcing is negative*
Yes, of course. We corrected the text.

"Radiative effects of the anthropogenic aerosol are negative and exceed 1 W m$^{-2}$..…"
*L160-165: How is aerosol microphysics calculated?*
We added the description of the aerosol block simulations in COSMO-ART:

"The simulations of reactive gaseous and particulate matter are based on the enhanced KAMM/DRAIS/MADEsoot/dust model (Riemer et al., 2003a; Vogel et al., 2006, Vogel et al., 2010). In MADEsoot (Modal Aerosol Dynamics Model for Europe extended by soot) all aerosol modes are represented by lognormal distributions. Five modes for Aitken and sub-micron aerosol particles include one pure soot mode, secondary particles consisting of sulphate, ammonium, nitrate, organic compounds (SOA), as well as the modes representing aged soot particles, consisting of sulphate, ammonium, nitrate, organic compounds, water, and soot. It also includes coarse particle mode, which contains additional anthropogenic emitted particles. All aerosol fractions are subject to coagulation and condensation following Binkowski and Shankar (1995), Whitby et al. (1991), Kerminen and Wexler (1994). Schell et al. (2001). Odum et al. (1996). The soot particles are directly emitted into the atmosphere. Coagulation and condensation are accounted for in transfer of soot from external into internal mixture. The Aitken and sub-micron particles are formed due to the aging process. For each mode prognostics equations for the number density and the mass concentration are solved numerically. The standard deviations are kept constant. Since the number densities of the coarse mode are small the inter-modal coagulation between the coarse mode and the other modes and the intra-modal coagulation of the coarse mode particles are both neglected. Additionally, the aerosol distributions are modified by the sedimentation, advection and turbulent diffusion processes. More details can be found in (Vogel et al., 2010). The resuspension of urban dust with stronger winds currently is not taken into account. "

*L175-178: Are there any natural emissions in the model, e.g., biogenic? Do you account for the resuspension of urban dust? How significant is it?*
There are biogenic emissions of hydrocarbons in the model. Figure 2 presents the distribution of all sources of aerosols gas precursors, including biogenic NMVOC, which shows the dominating urban source of these emissions.
In model simulations resuspension of urban dust with winds currently is not taken into account (urban dust is emitted as anthropogenic regardless of wind speed). It can be a source during strong winds, however, these situations are rare in spring in Moscow (mean wind velocity in spring is only V=2.6-2.8 ms-1 according to 60 years of measurements in Moscow (Chubarova et al., 2014, https://doi.org/10.3103/S106837391409005)).

We added in the description of the model the following sentence:
"The resuspension of urban dust with stronger winds currently is not taken into account."

*L187: In observations, one site is urban, and the another is in the suburb. It is not much coverage. Could you tell from the model simulations that these two locations represent urban and pristine conditions?*

Yes, these two locations represent urban and pristine conditions. The Figure 2 has been modified and the locations of both Moscow and Zvenigorod sites are shown now together with the fields of emissions. It is clearly seen that the maximum of anthropogenic emissions are over the MSU MO area and negligible emissions are seen over Zvenigorod site. Since we have prevailing westerlies, Zvenigorod station is not affected by the Moscow megacity anthropogenic aerosol emission, because pollutant plume from the city is transferred further to the east. MSU MO, on contrary, is affected by the anthropogenic emissions. In addition, we studied the specific conditions, when the effect of Moscow due to meteorological situations can affect Zvenigorod. The obtained differences in urban effects are not high (see the Discussion in 3.4).

In the text there is also the following information with reference to this Figure:

"Due to prevailing westerlies and location of the ZSS site far from local anthropogenic emissions (see Fig. 2) it can be characterized as a background pristine site."

The caption of Figure 2 has been modified accordingly:
**"Figure 2: Monthly mean emissions of aerosol gas-precursors, PM10 and BC emissions according to the TNO2010 inventory in April and May in Moscow region. The location of the Moscow State University Meteorological Observatory is shown by red circle and Zvenigorod site – by green circle."**

*L190-192: It is irrelevant to have instant observations at the same time. The averaging over time would give more reliable results.*

In the text it is written that "time difference between the two instant measurements in these sites is only 3 minutes." I guess this is the advantage to have the quasi simultaneous measurements of the initial data. Of course, then we averaged the data having the 1-hour resolution dataset, which has been further analyzed.

We added the following sentence in the text:

"Finally, all the data were combined in the 1-hour resolution dataset."

*L225-230: The definitions of the fine and coarse aerosol fractions came from AERONET. What are coarse and fine aerosols in the model? Could you explain this in the text?*
In the model there are different size ranges: Aitken mode, Accumulation mode and Coarse particle mode according to their standard definitions.
We added the information about the aerosol modes:
"Five modes for Aitken and sub-micron aerosol particles include one pure soot mode, secondary particles consisting of sulphate, ammonium, nitrate, organic compounds (SOA), as well as modes representing aged soot particles, consisting of sulphate, ammonium, nitrate, organic compounds, water, and soot. It also includes coarse particle mode, which contains additional anthropogenic emitted particles".
Please, find also the additional information about the model microphysics, which we have added above in the response.

*L260-265: I suppose sulfate is low because SO2 is low. What are the secondary aerosols in your observations and simulations?*
Yes, you are right, measured $SO_2$ concentrations are much lower than those, calculated using the TNO anthropogenic emission dataset. We mentioned this in the text.
In simulations, along with other aerosol modes, we have different kinds of secondary aerosols including sulphate, nitrate and secondary organic aerosol. Their description has been added in the text.
Unfortunately, currently there are no secondary aerosols measurements in Moscow. Concerning model simulations, secondary aerosols ratio in the total urban aerosol content is about 55% with maximum contribution for organic component (30%). This is mainly in accordance with the ranges provided in (Huang et al., 2014).

Huang, RJ., Zhang, Y., Bozzetti, C. *et al.* High secondary aerosol contribution to particulate pollution during haze events in China. *Nature* **514,** 218–222 (2014). https://doi.org/10.1038/nature13774

We added this information in the text.
"This may indicate the importance of secondary aerosol generation in the urban atmosphere of Moscow. According to the model simulations, secondary aerosols are about 55% in the total urban aerosol content, which is mainly in accordance to (Huang et al., 2014), with the maximum contribution of the organic component (30%)."

*L272-274: I suggest stronger winds more effectively generate coarse aerosols, like urban dust.*

Yes, there is a coarse aerosol content dependence on wind velocity. However, in the model currently there is no parameterization of urban dust resuspension. We have added in the text the following remark:

"A decrease in AEE and, correspondingly, the decrease in the fine AOD fraction with the increase in wind speed may be also associated with less effective fine mode aerosol generation due to better ventilation conditions and with possible more effective urban coarse aerosol mode dust resuspension at stronger winds (Amato et al., 2009, Hosiokangas et al. 2004)."

*L284-287: I would say that in unstable conditions, BC PM is more effectively dispersed vertically, the BC source is fixed, and the PM source gets stronger with stronger winds.*

We have edited this sentence by adding the possible effects of resuspension. However, it should be studied further in details in special experiments. The dust fraction is not large over our region.
The changes are the following:
The smaller $PM_{10}$ negative correlation with wind speed could be also explained by the effects of dust resuspension at stronger winds.

*L308-309: Do all Moscow power stations work on gas? What about SO2 emissions from traffic?*

Moscow power stations use gas as the main fuel, with other options for reserve fuels. Generally Moscow SO2 emissions are overestimated by TNO emission inventory as seen from our analysis. The Euro-5 motor fuel standard has been used in Moscow since 2016 that provides low SO2 emissions in the atmosphere.
We added this information in the updated text:

"In addition, Euro-5 motor fuel standard, which has been used in Moscow since 2016, provided low $SO_2$ emissions in the atmosphere."

*L369: "results" > "result"*
Done. Thank you.

*L400: Do anthropogenic emissions have a diurnal cycle?*
Yes, TNO inventory, which is used in the simulations, has a diurnal cycle with 1-hour resolution. We added this in the text.

"The one-hour resolution TNO2010 emission inventory has been developed using official reported emissions data by source category and combining them with other estimates, where needed (Kuenen et al., 2014)."

---

## Author Comment (AC2)

We are greatly appreciated the comments of the reviewer and his suggestions for including some additional analysis, which have been inserted in the new version of the text.

**Comment on acp-2022-83**

Anonymous Referee #2

Referee comment on "Columnar and surface urban aerosol in Moscow megacity according to measurements and simulations with COSMO-ART model" by Natalia Chubarova et al., Atmos. Chem. Phys. Discuss., https://doi.org/10.5194/acp-2022-83-RC2, 2022

This work focuses on the aerosol properties on the ground and in the atmospheric column, and their relationship with meteorological parameters, in the urban environment of the Moscow megacity. Data include ground measurements of PM10, BC and gaseous precursors, columnar aerosol parameters retrieved by AERONET data, as well as modeled data obtained by the application of the COSMO-ART model. Additional data obtained in an upwind clean background site at Zvenigorod Scientific Station where used for the estimation of the urban component of the studied aerosol parameters. Overall, this is a very interesting work, based on sound experimental and numerical methods. To my knowledge, there are few works on the aerosol pollution in the Moscow megacity, rendering this information more valuable. I believe the manuscript merits publication, after some minor revisions are done, as indicated in the specific comments below.

*A careful editing of the whole manuscript is needed in order to correct syntax errors (e.g. in lines 39-40, 59-61, 66-68, 70-75, among others).*

We have edited the whole text focusing on improving the style and on removing syntax errors. All changes are shown in green color.

*Line 121: Please specify the time resolution of the eBC measurements, in accordance with the other online in situ measurements you mention above.*

Aerosol equivalent BC (eBC) mass concentrations were measured with 1-minute resolution. We added this information in the text.

"Aerosol equivalent BC (eBC) mass concentrations were measured with 1-minute resolution using custom-made portable aethalometer (Popovicheva et al., 2017)."

*Figure 2: Is the red circle displayed in the maps the measurement site? Please clarify.*

Yes, sorry, it was not clearly explained. Yes, this is a mark for the MSU MO location. The caption has been changed. We also inserted a mark for the Zvenigorod site location. See the updated Figure in the Supplement.

[Figure]

**NO$_2$ (TNO2010, april)**

**NO$_2$ (TNO2010, may)**

kg/m$^2$*s$^{-1}$

kg/m$^2$*s$^{-1}$

**NO (TNO2010, april)**

**NO (TNO2010, may)**

kg/m$^2$*s$^{-1}$

kg/m$^2$*s$^{-1}$

**SO$_2$ (TNO2010, april)**

**SO$_2$ (TNO2010, may)**

kg/m$^2$*s$^{-1}$

kg/m$^2$*s$^{-1}$

**PM$_{10}$ (TNO2010, april)**

**PM$_{10}$ (TNO2010, may)**

kg/m$^2$*s$^{-1}$

kg/m$^2$*s$^{-1}$

[Figure]

[Figure]

**Figure 2: Monthly mean emissions of aerosol gas precursors, PM10 and BC emissions according to the TNO2010 inventory in April and May in Moscow region. The location of the Moscow State University Meteorological Observatory is shown by red circle and Zvenigorod site – by green circle.**

The caption has been changed in the following way:

**Figure 2: Monthly mean emissions of aerosol gas-precursors, PM10 and BC emissions according to the TNO2010 inventory in April and May in Moscow region. The location of the Moscow State University Meteorological Observatory is shown by red circle and Zvenigorod site – by green circle.**

*Also, please comment with respect to how representative are the 2010 emissions for the measurement period (2018 and 2019).*

Before the application of the TNO2010 emissions we have made the comparisons with the available data on the older TNO2003-2007 dataset emissions for aerosol retrievals. According to the earlier data we received much larger AODurb effect, while the TNO2010 emissions provided close results to measurements for the year of our previous study (2018) (0.055 and 0.023 respectively compared with measured value of 0.03). That is why for the new experiments we decided to use these TNO2010 inventory data. The new preliminary experiments for Moscow, which are currently going with modern new CAMS-2019 emissions, have shown also similar level of the simulated urban aerosol and the same problems with too high SO2 emissions (mean values over Moscow are 0.7e-09 kg*m-2*c-1 (TNO2010) and 1e-09 kg*m-2*c-1 (CAMS).). In addition, we should mention that the intensity of anthropogenic pollutant emissions should be adapted to be used in Chemical Transport Model, and this adaptation proves to have more impact on the simulation quality than the inventory updates. The newer dataset was used in the continued work, but for the study covered in this paper TNO2010 data proved to be appropriate, except for SO2 emissions, which is discussed in the text.

We added the following text on this account:
Testing the model estimates with the TNO2010 and TNO2003-2007 inventory datasets against observations provided much better agreement for urban aerosol, when TNO2010 was used ("Aerosol urban pollution and its effects on weather, regional climate and geochemical processes", 2020). This enables us to apply TNO2010 inventory in this study. The preliminary comparisons with the modern CAMS inventory dataset for 2019 also showed an agreement of the urban aerosol estimates.
*Lines 198-199 "We consider that our BC measurements in Moscow provide the BCurb component, whereas the black carbon is mainly formed and emitted in the urban environment (see Fig. 2).": The BC emissions are not depicted in Figure 2; please clarify*

*what you mean by citing Fig. 2. In any case, I agree that BC is mainly emitted in the urban environment.*

Sorry! Somehow BC emissions disappeared from Fig.2. Now they are included there.

[Figure]

**Figure 2: Monthly mean emissions of aerosol gas-precursors, ==PM10 and BC emissions== according to the TNO2010 inventory in April and May in Moscow region. ==The location of the Moscow State University Meteorological Observatory is shown by red circle and Zvenigorod site – by green circle.==**

*Table 1: Please specify that PM10 and BC measurements correspond to surface observations and not the total column of the atmosphere, as noted in the Table caption. Also, please correct "Confident interval at 0.05" to "Confidence interval at 0.05".*

Done.  The updated caption is the following:

**Table 1.** Statistics of hourly mean aerosol characteristics in the total column of the atmosphere ==and at surface== including aerosol optical depth (AOD at 500nm), Angstrom Extinction Exponent, Fine mode fraction at 500 nm, single scattering albedo (SSA) at 675nm, factor of asymmetry (ASY) for fine, coarse and total aerosol at 675nm, water vapor content (W, cm) and ==surface== mass concentrations of $PM_{10}$, BC, different aerosol gas-precursors, and $BC/PM_{10}$ ratio. Moscow. April-May, 2018-2019.

*Figure 4: Please note the units for all parameters presented. You may include this in the caption, if it is too complicated to include it on the axis.*

Done. We are not sure that it is important to mark, that AOD and IPD are unitless values.
We added missed units in the Caption:

**Figure 4: Daily means of AOD500, fine mode AOD fraction at 500 nm, $PM_{10}$ (==in mg m$^{-3}$==) and aerosol gas precursors (NOx, CHx) mass concentrations (in mg m$^{-3}$), $BC/PM_{10}$ ratio, water vapor content (in cm) and IPD index in 2018 (a) and 2019 (b).**

*Table 2 and related discussion: Most of the correlations discussed in Lines 260-287 are statistically significant but display a low correlation coefficient. In my opinion, a correlation coefficient below 0.5 does not really imply significant relationship between the two parameters. For example, the authors state "We obtained a statistically significant correlation of columnar AOD500 with surface PM10, and BC. A more pronounced dependence of both BC and PM10 with fine AOD500 mode could be explained by the fine mode BC composition and the predominant fraction of fine aerosol mode in PM10 in urban aerosol in Central and Northern Europe"; these correlations correspond to coefficients of 0.34 and 0.39 for BC with columnar AOD500 and fine AOD500 mode, respectively, which I*

*think are too low to show real correlation. A better correlation is observed for PM10 with AOD500; nevertheless, I don't see a difference between AOD 500 and fine AOD500 mode (correlation coefficient of 0.57 versus 0.58). Similarly, no correlation can my claimed for the fine AOD500 mode and SO2 concentrations, while only low correlations are observed for the other gaseous precursors.*

We agree with the reviewer that in some case our statements were too optimistic and correlations between some characteristics, of course, were not very high. Of course, a lot of other factors are also important, that makes the correlation low.  We have updated the text in the following way to smooth or remove these statements:

"A correlation matrix has been estimated for evaluating the relationship between different columnar and surface aerosol characteristics, aerosol gas precursors and meteorological parameters (Table 2). There is high correlation between AOD500 and fine AOD500 mode, which is dominant in Central  and East Europe (Logothetis et al., 2020). The prevailing fine aerosol mode fraction is also observed in $PM_{10}$ in urban conditions over Central and Northern Europe (see, for example, Fig. 10 in Wu and Boor, 2021). Relatively high correlation is detected between surface measurements of $PM_{10}$ with BC and gas aerosol precursors, except $SO_2$, which indicates the importance of these substances for aerosol formation. We also obtained a statistically significant, but not very high correlation of columnar AOD500 with surface $PM_{10}$, and BC. Fine AOD500 mode has slightly higher correlation with BC, which could be explained by the fine mode BC composition (Bond et al., 2013). The importance of secondary urban aerosol in columnar fine mode AOD500 (Dubovik et al., 2002) has been also proved by a statistically significant correlation between fine AOD500 mode and aerosol gas precursors ($NO_x$, $SO_2$, CHx), however, the correlation coefficients are not high due to complexity of the chemical and meteorological processes. "

*I believe a more fruitful discussion may be based on the correlation analysis graphs (e.g. Figures 5 and 6), where interesting observations can be made (as in Lines 289-309). Please also consider revising the corresponding comments in the Conclusions section.*

Concerning Fig5, we suppose that the correlation analysis would not be very indicative. We made some changes in the following way:
"A more detailed analysis of the relationship between AOD500 and $PM_{10}$ surface mass concentrations shown in Fig. 5a demonstrates that along with the existence of general not very high correlation (see Table 2) there is a split into two types of dependences at a point of bifurcation of $PM_{10}$ ~0.05 $mg\,m^{-3}$ . A weaker AOD500 dependence versus $PM_{10}$ characterizes the accumulation of $PM_{10}$ only in the low layer (due to local emission sources near the surface) in the absence of the pronounced AOD500 increase with many cases at IPD=1, relating to the low intensity of particle dispersion. A more pronounced dependence between AOD500 and $PM_{10}$ is associated with the influence of air mass advection, when the concentration of surface particles increases simultaneously with AOD500. In this case only few cases at IPD=1 are observed (Fig. 5a). The increase in $PM_{10}$ is also connected with a significant increase in fine mode AOD500 fraction and the total absence of its low values at high $PM_{10}$ levels (Fig. 5b). The existence of these two dependences may explain not very high correlation between AOD500 and $PM_{10}$ for the whole dataset.

In Discussion and in Conclusions the text has been changed/removed.

In Discussion:
We obtained high correlation between AOD500 and fine AOD500 mode, which  was typical for Central  and East Europe (Logothetis et al., 2020). Fine aerosol mode fraction is also dominating in $PM_{10}$ in urban regions of Central and Northern Europe (Wu and Boor, 2021). We found relatively high correlation between surface measurements of $PM_{10}$ and BC with gas aerosol precursors, except $SO_2$.

In Conclusions:

We found the predominance of fine AOD500 mode in AOD500 and a statistically significant, though not very high correlation, between columnar AOD500 and surface $PM_{10}$ mass concentrations due to splitting their dependence in two different ones. Relatively high correlation between surface measurements of $PM_{10}$ and BC are observed with aerosol gas precursors, except $SO_2$.

Concerning Fig.6: we have added the analysis of correlation for different IPD regimes in the following way: "Note, that the account of IPD can additionally increase correlation between BC and $PM_{10}$ (R=0.94 for IPD=1, R=0.81 for IPD=3, compared with R=0.64 for the whole dataset). Similar, however, smaller increase in correlation is observed after the IPD account in relationship between BC and $NO_2$ (R= 0.74 for IPD=1, R=0.85 for IPD=3 compared with R=0.7 – for the whole dataset)."

*Line 318: Please correct "solar elevations" to "solar radiation".*

Done. Thanks.

*Figure 8: I would suggest to include also the % of the urban component with respect to total variable. In my opinion, the absolute values of the urban component of the different parameters do not show clearly the impact of the city on local/regional air quality.*

Thank you for the suggestion. We added the ratio of AODurb/Total AOD at 500 nm and some description of this quantity. The absolute values are also important, since they demonstrate in the first approximation, radiative effect of urban aerosol, because AOD is the characteristics of solar irradiance attenuation.

The text has been changed in the following way:

The $AOD_{urb}$/AOD ratio at 500nm comprises about 19%. No statistically significant difference in the coarse AOD mode was found between Moscow and clean unpolluted site.

[Figure]

**"Figure 8: Annual mean urban components of different aerosol parameters - $AOD_{urb}$ at 550nm and at 500 nm, Fine Mode $AOD_{urb}$, Coarse Mode $AOD_{urb}$, urban component of the Angstrom Extinction Exponent $AEE_{urb}$ and AODurb/AOD500 ratio with confidence intervals at 0.05 significance level. The inset shows the mean $AOD_{urb}$ spectral dependence. Comment: we show the AOD at two wavelengths to provide more convenient comparisons with the CIMEL sun-photometer observations (AOD at 500 nm) and model results (AOD at 550 nm). Moscow. 2006-2020."**

*Figure 9 caption: "Figure 9: (a) - Time series of AOD at 550 nm simulated using direct observations of AOD 915 at 500nm and AEE at 440-870nm, and the AODurb component according to measurements and modelling in 2018 (left upper panel) and 2019 (right upper panel)": The caption is not so clear. The simulated AOD at 550 nm corresponds to the total AOD or maybe the AODurb component?*

Sorry, the caption of this Figure was unclear. We consider both urban AODs and total AOD taken from measurements. The updated caption is the following:

Figure 9: (a) - Time series of AOD550 from observations (AOD Moscow (obs)), and its urban components from observations (AODurb (obs)) and modelling (AODurb(mod)) in 2018 (left upper panel) and 2019 (right upper panel); (b) – Time series of $PM_{10}$ from observations ($PM_{10}$ Moscow (obs), in mg m$^{-3}$), and its urban components (in mg m$^{-3}$) from observations ($PM_{10urb}$(obs)), and modelling, ($PM_{10urb}$(mod)). 2018.

*Table 3: I don't understand how the PM10 model values were obtained. According to the Methods section (and in particular Lines 176-178), only the anthropogenic component of the surface mass concentrations of PM10 was simulated. The same question holds for BC model data presented in Table 3. In addition, how do the authors obtain the BC value used in the BCurb/BC ratio mentioned in Line 367?*

Sorry it was not well written in the text. We made changes in the section, where the methods are described. Yes, no aerosol is presented at boundary layers, but over our territory we have small biogenic natural emissions. This means that when we simulate PM10 and AOD, we should extract their small content from total content for having comparisons with measured urban component of aerosol. Now we added the description in the text in the Section 2.1.2:

Urban aerosol sources according to the TNO2010 include direct emission of particulate matter of undefined composition (as dust fraction) and soot, as well as the gas-aerosol precursors of sulphate, organic, and nitrate secondary aerosols. The model also accounts for relatively small natural biogenic emissions of non-methane volatile organic compounds from the Global Land Cover 2000 project, which are the gas precursors of organic aerosol.

*Lines 371-375 "We analyzed if there is a relationship between urban aerosol component and the total aerosol content. Figure 10 presents the dependence of model and measured AODurb on total AOD according to the MSU MO measurements, and the dependence of PM10urb on PM10. There is a positive correlation of urban aerosol component for AOD and PM10 with total AOD and PM10.":*
*I think there are some issues with respect to the discussion of Figure 10.*
*For one, since the urban component is part of the total variable, a good positive correlation between these two does not clearly imply the simultaneous formation of natural and anthropogenic aerosol; It may be also related to meteorological conditions favouring the accumulation of pollutants. Also, a good correlation as the one displayed in Figure 10(c) suggests a constant PM10urb/PM10 ratio, so a constant % contribution of the urban component.*
*In the case of AOD, I don't think the AODurb observations (Figure 10(a)) demonstrate a correlation with the total AOD. For the AODurb model (Figure 10(b)), there seems to be a positive correlation for higher AOD values (above 0.2). I think a better picture could be obtained if the authors plotted the ratio of PM10urb/PM10 (and AODurb/AOD) over the PM10 (and AOD). It would be interesting to comment if for higher AOD values, the anthropogenic component contributes more. For PM10, this does not seem to be the case.*

Thank you for the suggestion. We added the ratios on the plots with the absolute values. You are right. The PM10 urban component is quite stable according to measurements. It is more complicated figure for AOD. We made the necessary changes and added the following text and updated Fig.10:

Urban aerosol may have relationship with natural aerosol, since they both are determined by chemical composition of the atmosphere and meteorological conditions. For evaluating their relations, we analyzed the dependences between the urban aerosol component and its total amount. Figure 10a,b presents model and measured AOD$_{urb}$ and AOD$_{urb}$/AOD550 ratio as a function of total AOD550 according to the MSU MO measurements. There are large

variations in AOD$_{urb}$ obtained from measurements and modelling. According to model results, there is a slight positive AOD$_{urb}$ increase at AOD550>0.2. The absence of the dependence for measured AOD$_{urb}$ versus AOD at 550 nm (see Fig. 10a) can be observed due to a significant contribution of the advection of natural aerosol with high AOD. Figure 10c,d presents similar dependencies but for PM$_{10urb}$ and PM$_{10urb}$/PM$_{10}$ ratio as a function of the observed PM$_{10}$ in Moscow. At surface layer, a positive correlation dependence between PM$_{10urb}$ and PM$_{10}$ is more pronounced, especially that obtained from observations. This can be explained by higher concentrations of aerosol gas precursors both of urban and natural origin, which, in turn, have high correlations with PM$_{10}$ (see Table 2).

The analysis of AOD$_{urb}$/AOD550 and PM$_{10urb}$/PM$_{10}$ ratios has revealed a tendency to decrease at high aerosol content. This may mean that large aerosol content in Moscow is observed due to advection, while the contribution of urban aerosol (higher than 50-100%) is important at relatively small aerosol level of about AOD=0.1-0.2 or PM$_{10}$ <0.04 mgm$^{-3}$.

[Figure]

(a)                                                                (b)

(c)                                                                (d)

**Figure 10: Measured (a,c) and model (b,d) urban component of aerosol optical depth at 550 nm (AOD$_{urb}$), urban PM$_{10}$ mass concentration (PM$_{10urb}$, in mg m$^{-3}$) and their ratios to the observed total AOD550 and PM$_{10}$ as a function of the observed total AOD550 (a,b) and PM$_{10}$ (c,d) in Moscow (MSU MO). For consistency reason we show only quasi-simultaneous AOD and PM$_{10}$ measurements during the daytime period with AOD observations. The cases affected by biomass burning aerosol were excluded. Clear sky conditions.**

*Lines 376-377 "This may be also accompanied by higher concentrations of aerosol gas precursors both of urban and natural origin, which, in turn, have high correlations with PM10 and AOD according to Table 2.":*
*According to Table 2, I don't think the authors can claim a high correlation between gas precursors and AOD (R between 0.17-0.34). For PM10, I agree that the correlation coefficients show high correlations (at least for NO2 and CHx).*

We have made following changes removing AOD from this paragraph:

This can be explained by higher concentrations of aerosol gas precursors both of urban and natural origin, which, in turn, have high correlations with $PM_{10}$ (see Table 2).

*Lines 395-397 "Both simulated and measured PM10urb values have a pronounced dependence on IPD with higher PM10urb at lower level of intensity of particle dispersion. Note, that the influence of the intensity of particle dispersion on AODurb is not observed.":*
*I don't understand how we can observe the dependence of the PM10 urban components on IPD, based on Figure 11. The relationship between the data obtained from model and observations varies, depending on the IPD value; but the PM10urb values display a wide range in all 3 cases of IPD.*

You are right, of course. It was a wrong interpretation of data. Only for the simulated PM10 we have the dependence on IPD , but no simultaneous effects in both measured and modeled urban component have been obtained. This part of the text has been removed.

*Figure 12: To my understanding, IPD will also vary during the day, so I am not sure how different diurnal cycles may be calculated for different IPD values. I would expect to see the diurnal cycle of the urban component (based on model and measurements), along with the emissions (when available) and the IPD value. The day-time and night-time meteorology and atmospheric conditions affect the IPD levels, so I am not sure what is the meaning of selecting the IPD values. Did the authors group the days by a daily average IPD value and then calculate the average diurnal cycle for each group (for mean 24hr IPD =1, 2 or 3)?*

Fig.12 was made as a composite of hourly values at different IPD values. We have changed the caption and made necessary changes in the text. We guess this is a way to show the effect of the influence of the intensity of particle dispersion on urban fraction of different aerosol characteristics.

"Figure 12 shows the composite diurnal cycles of AOD, $PM_{10}$ and BC at different IPD, as well as the primary emissions of black carbon and $PM_{10}$ according to TNO2010 inventory. In general, there are noticeable diurnal changes of model and experimental data at the surface layer, which has some specific features depending on IPD. One can see the accumulation of $PM_{10}$ and BC at night below the inversion layer in the stable atmosphere, which is characterized by IPD=1. Note, that during daytime (from 10 to 17 h) the conditions with IPD=1 were never recorded, because of warming up the surface and the amplification of convection.
As for columnar AOD characteristic, there is no evident diurnal cycle of measured $AOD_{urb}$ during daylight hours, however, model $AOD_{urb}$ values demonstrate a small increase at night, especially, in conditions with IPD=1. Figure 12b,c shows a noticeable dependence of $BC_{urb}$ and $PM_{10urb}$ on IPD index, especially for night and early morning conditions. Elevated values of the surface urban aerosol at night in conditions with IPD=1 reach 30-40 $\mu g\,m^{-3}$ for $PM_{10urb}$, and 3-3.5 $\mu g\,m^{-3}$ - for $BC_{urb}$. "
The caption is the following:

**Figure 12:** The composites of the  diurnal cycle of AOD $_{urb}$(a), PM$_{10\ urb}$ (b) and BC urb mass concentrations (c) according to measurements and modelling for different conditions of the intensity of particle dispersion (IPD). For PM$_{10}$ and BC the diurnal cycle of their emissions is also shown. The confidence intervals were calculated at 0.05 significance level. Moscow.

---

## Author Response (AR2)

I would like to thank the Editor for the suggestions, how to improve the English style of the text.

Here is the list of changes. We also made many other improvements after consulting with the English speaking specialist. All changes are shown in the file with tracking marks.

Line 22: "relatively low BC/PM10 ratio (for urban regions) of 4.7%"

Done

- Line 29: consider using microgram per m3 instead of milligram per m3 as PM10 units

Done in this and in similar places.

- Line 33: "although there was a slight increase in..."

Done

- Line 38: "compensating for the increase..."

Done

- Line 41: "(Seinfeld and Pandis, 2016)". There are also other formatting issues with the references.

These can be fixed in the post acceptance editing stage.

Done. We also made some other corrections in the references

- Line 77: "for reliable estimations of"

Done

- Line 89: This sentence does not make sense. Moscow is not in North America. Perhaps rephrase to "Air pollutant levels in Moscow are consistent with those in other megacities in Europe and North America."

Done.

- Line 91-94: Also an awkward sentence. Rephrase based on the previous suggestion.

New variant is the following:

The levels of BC concentration are similar to those in European cities, that indicates the comparable impacts of major urban sources such as traffic with intensive implementation of modern environmental requirements, heating power plants, and manufacturing industries (Popovicheva et al., 2020, Popovicheva et al., 2022).

- Line 116: replace "in operation by" with "operated by"

Done

- Line 117: should it be "MosEcoMonitoring"?

No. The initial variant is right.

- Line 137: reference is not complete

Since this is a book, the reference was changed to (Chubarova, 2020)

- Line 144: unclear what "It" in "It is calculated by..." is referring to

Done.

New variant is "This index is calculated"

- Line 211: "we mainly consider the simulation of the anthropogenic..."

Done

- Line 218: "over the Moscow megacity"

Done

- Line 222: "In a similar manner..."

Done

- Line 257: "is in agreement with"

Done

- Line 275: This sentence "Note, that over remote areas in Paerne (Switzerland) and Reunion Island (France) BC concentrations comprise 0.4-0.5 μgm-3 according to (Gerich et al., 2011, Bhugwant and Brémaud, 2001)" contain multiple grammatical errors. Please correct.

This sentence has been changed to:

Over remote unpolluted areas, however, BC concentrations are about 0.4-0.5 $\mu gm^{-3}$ (Bhugwant and Brémaud, 2001; Herich et al., 2011).

Line 288: remove "an"

Done

Line 295: grammatical error

Removed.